# Subsurface Scattering for 3D Gaussian Splatting

**Jan-Niklas Dihlmann**     **Arjun Majumdar**     **Andreas Engelhardt**     **Raphael Braun**

**Hendrik P.A. Lensch**

University of Tübingen

## Abstract

3D reconstruction and relighting of objects made from scattering materials present a significant challenge due to the complex light transport beneath the surface. 3D Gaussian Splatting introduced high-quality novel view synthesis at real-time speeds. While 3D Gaussians efficiently approximate an object's surface, they fail to capture the volumetric properties of subsurface scattering. We propose a framework for optimizing an object's shape together with the radiance transfer field given multi-view OLAT (one light at a time) data. Our method decomposes the scene into an explicit surface represented as 3D Gaussians, with a spatially varying BRDF, and an implicit volumetric representation of the scattering component. A learned incident light field accounts for shadowing. We optimize all parameters jointly via ray-traced differentiable rendering. Our approach enables material editing, relighting and novel view synthesis at interactive rates. We show successful application on synthetic data and introduce a newly acquired multi-view multi-light dataset of objects in a light-stage setup. Compared to previous work we achieve comparable or better results at a fraction of optimization and rendering time while enabling detailed control over material attributes. Project page: https://sss.jdihlmann.com/

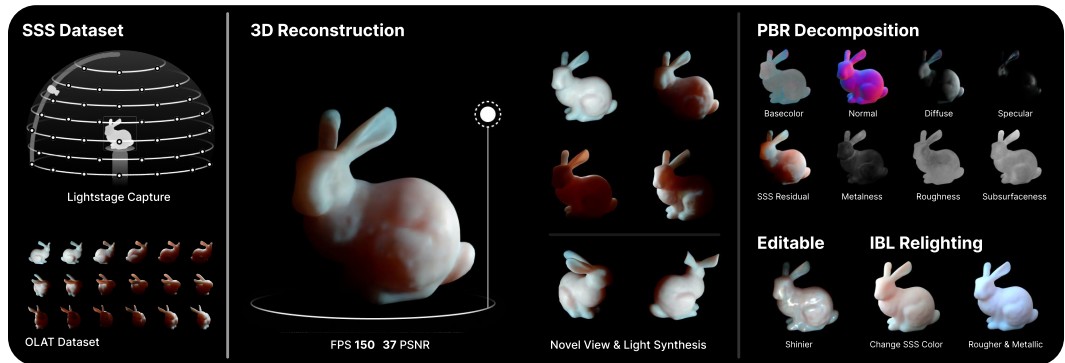

Figure 1: **SSS GS** – We propose photorealistic real-time relighting and novel view synthesis of subsurface scattering objects. We learn to reconstruct the shape and translucent appearance of an object within the Gaussian Splatting framework. To do so we leverage our newly created multi-view multi-light dataset of synthetic and real-world objects acquired in a light-stage setup. The object is decomposed in a PBR fashion allowing for easy material editing and relighting. For a trailer visit our project page at https://sss.jdihlmann.com/.

## 1 Introduction

Subsurface scattering (SSS) is an important aspect of our visual reality and therefore an indispensable part of realistic rendering. It is the process by which light penetrates a surface and is scattered

38th Conference on Neural Information Processing Systems (NeurIPS 2024).

beneath it before being reflected back out. This phenomenon is responsible for the soft and diffuse appearance of materials such as wax, marble, skin and many other organic substances. Modeling SSS is challenging because it requires capturing the complex interactions between light and matter between different points on the surface. Approximating SSS by a simple surface reflection model where the light is reflected directly at the point of incidence typically leads to unnatural appearance of those objects. As a result, efforts in computer graphics to explicitly model SSS are either computationally expensive or approximate for interactive rendering. Additionally, capturing the complexity of spatially varying real-world scattering properties of an object using conventional computer vision techniques is difficult as they are often focused on reconstructing only visible surfaces and their BRDF.

In recent years, modeling SSS using neural networks has been a topic of interest, e.g. [36, 29] use Neural SSS materials as part of a Monte Carlo global illumination rendering pipeline. These methods are trained in a supervised fashion using synthetic datasets. In contrast, Neural Radiance Fields (NeRFs) [27, 49, 25] can learn the volumetric properties of SSS under varying lighting conditions implicitly and achieve photorealistic novel view synthesis and relighting. However, NeRFs are slow in training and inference due to the volumetric rendering requiring large or multiple MLPs to be evaluated for many point samples along each ray. Recently, 3D Gaussian Splatting (3D GS) [18] has been introduced as a method for 3D reconstruction with high-quality novel view synthesis. It achieves real-time speeds by avoiding costly volume rendering, which, however, also limits the representation of volumetric effects. In this work, we introduce subsurface scattering to the 3D GS framework.

Specifically, we propose the first method based on 3D GS for capturing detailed SSS effects of single objects, allowing for rendering and relighting in real-time.

At the core, we propose a hybrid representation that extends 3D Gaussian Splatting with PBR material parameters [10] and deferred shading [12] with a light-weight residual prediction network to learn a subsurface scattering (SSS) shading component not modeled by the surface shader. We constrain the network predicting the outgoing SSS radiance by jointly predicting the incident radiance used for the PBR rendering step enforcing a neural representation of the local and global light transport in the scene. To overcome the inherent resolution limit of 3D Gaussians we perform the shading in image space. This improves the representation of specularity in particular.

We further introduce a newly acquired OLAT (one-light-at-a-time) dataset of SSS objects. It comprises object-centric 360° multi-view image collections of synthetic objects rendered in Blender using the Cycles PBR renderer [8] as well as real-world examples we acquired using a light stage and motorized camera.

Our method provides object based decomposition for PBR material components and SSS effects, which allows for editing and novel material synthesis. We show that our method has improved training time and faster rendering speed compared to previous SSS approaches based on NeRFs while achieving comparable or better results.

Possible applications include • Medical Imaging and Visualization for Tissue Rendering and Surgical Simulation [35, 40] • Entertainment for visual effects and animation [30] • VR and AR by providing a more realistic and immersive experience.

## 2 Related Work

**Scene Representation**    Neural Radiance Fields (NeRF) introduced by Mildenhall et al. [27], have started a trend in synthesizing novel views of complex 3D scenes with high fidelity, by representing the scene as a continuous 5D function using neural networks. While NeRFs have been widely adopted, they are computationally expensive due to the volumetric evaluation. There have been several works to accelerate NeRFs [28, 38, 33] one of which is KiloNeRF [31] that uses multiple small MLPs to represent different parts of the scene. Kerbl et al. [18] introduced representing the scene as a set of explicit learnable 3D Gaussians. The proposed method 3D Gaussian Splatting (3D GS) is more efficient than NeRFs due to splatting i.e. rasterization of the Gaussians. However, both these representations are limited to static scene representations and do not support relighting or material decomposition.

**Material Decomposition and Relighting**    There have been prior works accomplishing 3D shape and material reconstruction for relighting: NeRD [3] was one of the first works to extend NeRF [27]

to decompose the input images into shape, BRDF and illumination for relighting. NeRFactor [47] and NeRV [7] aim for similar goals with slightly different network architectures. Neural-PIL [4] added a split-sum pre-integrated illumination network for more accurate and faster optimization. Likewise, NVDiffrec [13] use a pre-integrated illumination representation to optimize materials together with the object's shape.

In the realm of Gaussian Splatting, there have been several works focusing on relighting and BRDF decomposition of static scenes [10, 17, 23]. With Relightable 3D Gaussians (R3DGS) [10], the authors decompose the scene into explicit metallic, roughness, color, and normal components, which can be composed and relit in real-time. The method models light with a neural incident light field [44] and a local learnable representaion as Spherical Harmonics. The illumination and shading is optimized per scene with differentiable rendering. Gaussian Shader [17] uses a similar decomposition but predicts the environmental illumination to not only model diffuse but also reflective surfaces. Further, works have utilized deffered shading [12] with Gaussian Splatting [39, 23, 19] to improve specular reflections, by focusing on the blending and propagation of normal directions between overlapping 3D Gaussians. Additionaly DeferredGS [39] train a SDF in parallel to improve the surface geometry. We identify that the areas of the 3D Gaussians are a key factor for the representation of high frequency details in the scene and propose deferred shading to improve the specular reflections without the need of normal optimization or additional SDF training. We base our work of R3DGS [10] and augment the reflectance calculation to accommodate for subsurface scattering.

**Subsurface Scattering**   There have been a multitude of prior works in SSS. Jensen et al. [16] presents a practical model for subsurface light transport in translucent materials that captures effects beyond BRDF models. Donner et al. [9] further explore light diffusion in multi-layered translucent materials using a variant of the Kubelka-Munk theory. Lensch et al. [22] introduce a rendering method for translucent diffuse objects, in which viewpoint and illumination can be modified at interactive rates. and filtered using the precomputed kernels. Vicini et al. [37] introduce a new shape-adaptive BSSRDF model that is based on a conditional variational autoencoder which learns to sample from a reference distribution produced by a brute-force volumetric path tracer. The distribution is conditional on both material properties and a set of features characterizing geometric variations in the neighborhood of the incident location. Zheng et al.[48] learn neural representations for participating media with a complete simulation of global illumination. They estimate direct illumination via ray tracing and compute indirect illumination with Spherical Harmonics. Zhu et al. [49] propose a novel framework for learning the radiance transfer field via volume rendering and utilizing various appearance cues to refine the geometry end-to-end. They extend relighting and reconstruction prospects to tackle a wider range of materials in a data-driven manner. They use a NeRF-like architecture to represent subsurface scattering based on known incident light directions from controlled acquisition. Similarly, Neural Radiance Transfer Fields [25] use one light at a time (OLAT) data as supervision to learn global light transport. This data representation is very similar to ours, however, their dataset of translucent objects has not been released so far. Due to using a NeRF-only approach they lack editing capabilities and have longer run times. Object-Centric Neural Scattering Functions (OSF) [46] introduce a framework for representing and rendering objects under varying lighting conditions and from arbitrary viewpoints using object-centric neural scattering functions. OSFs allow for flexible composition of scenes with multiple objects, each maintaining realistic interactions with light and shadows but they also suffer from long training time, even their variant based on KiloNeRF [31]. Further, in the realm of Gaussian Splatting, there have been works on human avatars incorperating some modeling of subsurface scattering with Spherical Harmonics [32].

Our method merges the implicit MLP-based shader representation for SSS with the efficiency of 3D GS. Although this is not the first time that 3D GS has been augmented with an implicit component, e.g. previously used to simulate view-dependent effects [41, 26], ours is the first to adapt it for modeling SSS.

## 3   Method

Our goal is to reconstruct photorealistic 3D objects with strong subsurface scattering (SSS) effects from multi-view, multi-light image sets and to render them in real-time. We propose a novel method that extends 3D Gaussian Splatting (3D GS) with an explicit surface appearance model and combines it with an implicit SSS model (Fig. 2).

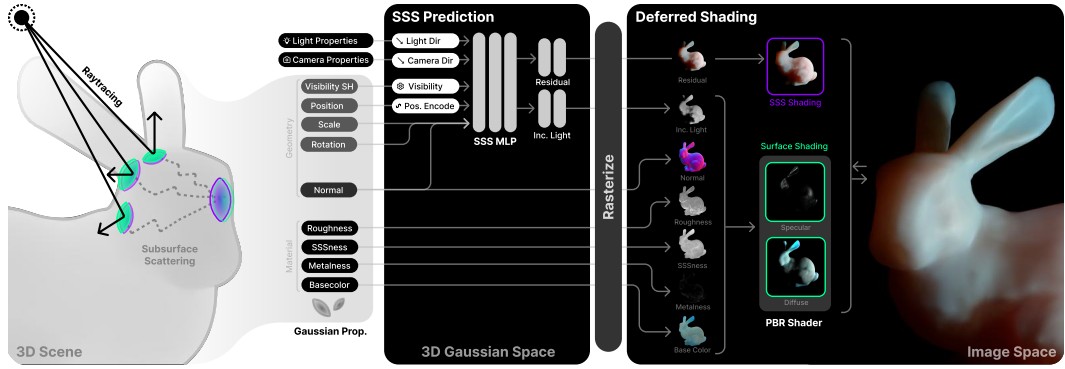

Figure 2: **Subsurface Scattering Pipeline** - Our method implicitly models the subsurface scattering appearance of an object and combines it with an explicit surface appearance model. The object is represented as a set of 3D Gaussians, consisting of geometry and appearance properties. We ultilize a small MLP to evaluate the subsurface scattering residual given the view and light direction and a subset of properties for each Gaussian. Further, we evaluate the incident light for each Gaussian as a joint task within the same MLP given the visibility supervised by ray-tracing. Based on the computed properties we accumulate and rasterize each property on the image plane in a deferred shading pipeline. We evaluate the diffuse and specular color with a BRDF model for every pixel in image space and combine it with the SSS residual to get the final color of the object.

## 3.1 Background

**3D Gaussian Splatting**   represents scene geometry and appearance using a set of 3D Gaussians [18]. Rasterizing these Gaussians with an efficiently designed splatting technique allows for fast 3D reconstruction and novel view synthesis. A single 3D Gaussian is defined by its mean $\mu$, i.e. the center position in 3D space, and the covariance matrix $\Sigma$ expressing its rotation and scale following the Gaussian function

$$G(x) = \exp\left(-\frac{1}{2}(x-\mu)^{\top}\Sigma^{-1}(x-\mu)\right),$$  (1)

Further, each Gaussian is associated with a color, modeled as Spherical Harmonics (SH) coefficients $c_i$, to represent view-dependent effects and an opacity $o$ for transparency. Consequentially, a set of 3D Gaussians is defined as $\mathcal{G} = \{(\mu_i, \Sigma_i, c_i, o_i)\}$. Rendering a scene represented by 3D Gaussians is done in the first step by projecting the Gaussians onto the image plane [50], where $J$ is the Jacobian of the affine approximation of the projective transformation and $W$ is the viewing transformation matrix. The covariance matrix is transformed as follows:

$$\Sigma' = JW\Sigma W^T J^T$$  (2)

The second step is to accumulate and rasterize the Gaussians onto the image plane, which is done by alpha blending the splatted colors and opacities of the Gaussians. The final color $C$ for pixel $(u, v)$ is given by the summation of the set of layered sequenced Gaussians $\mathcal{G}_i$ contributing to the pixel:

$$C_{u,v} = \sum_i \left(\prod_{j=1}^{i-1}(1-\alpha_j)\right)\alpha_i c_i$$  (3)

The $\alpha$ term is constructed by multiplying the opacity $o$ with the 2D covariance contribution $\Sigma'$ at a given pixel position. To facilitate optimization the covariance $\Sigma$ is parameterized as a 3D vector for scale $s$ and a unit quaternion $q$ for rotation.

**Relightable 3D Gaussians**   In Relightable 3D Gaussians (R3D GS) [10] the authors decompose the appearance of 3D Gaussians into explicit material properties, which can be relit in real-time. The method models light with a global neural incident light field [45] and a local learnable incident light representation based on SHs. The illumination and shading are optimized for a static scene with differentiable rendering.

Each Gaussian receives additional physically based rendering (PBR) parameters, such as a basecolor $b$, a roughness $r$, a metalness $m$, and a normal $n$. The approach adopts the Disney BRDF model [6] with the diffuse and specular terms as follows:

$$f_{\text{diffuse}} = \frac{1-m}{\pi}b, \quad \text{and} \quad f_{\text{specular}}(\omega_o, \omega_i) = \frac{D(h,r)F(\omega_o, h, b, m)G(\omega_o, \omega_i, h, r)}{(\omega_o \cdot n)(\omega_i \cdot n)}.$$  (4)

Here $D$ is the microfacet distribution function, $F$ is the Fresnel term, $G$ is the geometry term and $h$ is the half vector between the view and light direction. The method first optimizes the geometry of the scene and then the shading parameters such as the incident light field and the PBR parameters. Further, the normal is derived from the geometry based on the depth in the scene. The authors additionally build efficient ray-tracing for 3D GS to supervise a learnable visibility SH term $v$ per Gaussian, which guides the incident light field optimization.

While showing impressive relighting results on a variety of objects, it fails for translucent objects featuring SSS. The method is limited to learning from scenes with a single static illumination setting. Consequently, as SSS is not explicitly modeled, the SSS effect is baked-in into the basecolor parameter, preventing SSS effects from being correctly rendered during relighting (Fig. 5).

## 3.2 Subsurface Scattering for 3D Gaussian Splatting

By building upon the 3D GS framework including R3D GS, our approach extends it to capture the SSS effects of objects. An implicit neural SSS model is jointly trained with the explicit surface BRDF model. We utilize a global neural network to estimate the SSS effects for each Gaussian in the scene and therefore approximate the internal light transport. At the same time, the neural network also takes care of the incident illumination on the object including local visibility to allow for fast evaluation.

**SSS Modeling**  Our core contribution is the modeling of SSS effects in the 3D Gaussian representation. For photorealistic rendering, the internal scattering of light can be modeled with a BSSRDF [15] as simulated in the rendering equation as follows:

$$L_{\text{out}}(\mathbf{x}_{\text{out}}, \omega_{\text{out}}) = \int_A \int_\Omega f_{\text{sss}}(\mathbf{x}_{\text{in}}, \mathbf{x}_{\text{out}}, \omega_{\text{in}}, \omega_{\text{out}}) L_{\text{in}}(\mathbf{x}_{\text{in}}, \omega_{\text{in}})(\omega_{\text{in}} \cdot n)\, \mathrm{d}\omega_{\text{in}}\, \mathrm{d}x_{\text{in}}, \tag{5}$$

Here $f_{\text{sss}}$ is the BSSRDF, $L$ is the radiance, $\mathbf{x}$ is the position, $\omega$ are the directions, $A$ is the surface area of the object, and $\Omega$ is the hemisphere at $\mathbf{x}_{\text{out}}$. The light transport through the object is sketched on the left side of Figure 2. To exactly evaluate the scattered radiance due to the BSSRDF one would need to integrate all incoming light directions over the entire surface, which is expensive.

Rather than modeling the BSSRDF directly or explicitly evaluating the light transport by integrating over the surface we propose an implicit SSS shader. For a small surface area, in our case a 3D Gaussian, a neural network learns to estimate the outgoing radiance due to SSS when the entire scene is illuminated from a single direction. This global implicit network (Fig. 2) is formulated as follows:

$$(L_{\text{out (SSS)}}, \bar{L}_{\text{in}}) = f_\Theta(\mu, \Sigma, n, \omega_{\text{in}}, \omega_{\text{out}}, v), \tag{6}$$

where $f_\Theta$ is the neural network, $\mu$ and $\Sigma$ are the mean and covariance of a 3D Gaussian, $n$ is the normal, $\omega_{\text{in}}$ and $\omega_{\text{out}}$ are the incident and outgoing light directions, and $v$ is the shadowing term (see below). We apply a Fourier encoding to $\mu$ as in [27] which is omitted in Equation 6. The network is not only estimating the outgoing SSS radiance $L_{\text{out (SSS)}}$ but at the same time the incident light $\bar{L}_{\text{in}}$ is predicted which will later be used to evaluate the direct reflection at $\mu$. We want to note that $\bar{L}_{\text{in}}$ is a prediction of our model that is close to the true physical quantity $L_{\text{in}}$ but might be slightly offset to compensate for limitations in the model. By forcing the network to predict the incident as well as the outgoing SSS radiance we let the network implicitly learn about the local and global light transport in the scene. $L_{\text{out (SSS)}}$ and $\bar{L}_{\text{in}}$ predictions share the same MLP but apply separate output heads (Fig. 2).

Besides this general modeling of the outgoing radiance, we introduce a local parameter $sss$ per 3D Gaussian to control the ratio of SSS vs. direct reflection.

In order to constrain the network to predict physically plausible results, we combine it with the previously described BRDF model and optimize it jointly with the incident light field prediction. We formulate the combination of the BRDF and the SSS network as follows:

$$L_{\text{out}} = sss \cdot L_{\text{out (SSS)}} + (1 - sss) \cdot (f_{\text{specular}} + f_{\text{diffuse}}) \cdot \bar{L}_{\text{in}} \cdot (n \cdot \omega_{\text{in}}), \tag{7}$$

where $sss$ is subsurfaceness, which is optimized during training to balance the direct reflection and the SSS effects as formulated in OpenPBR [1].

**Incident Light**  We assume a single light source at a time (OLAT) setup, where the position of the light source is known. Similar to [10] we trace rays using a BVH to quickly estimate the visibility for

each Gaussian and learn this as a per Gaussian SH visibility term $v$. As explained above we model the incident light field with a neural network that takes the scene geometry, the light positions and the visibility into account. This network is jointly evaluated with our SSS radiance. We optimize the incident light field as one additional component of our differentiable rendering pipeline.

**Per-Pixel Deferred Shading** Instead of the direct illumination model of the R3DGS [10] we introduce a deferred shading [12] to capture sharper highlights. We noticed that computing the BRDF model in Gaussian space is insufficient to capture high-frequency details such as specular highlights. This is because some Gaussians represent a large surface area as seen in Figure 5. A single point-wise evaluation of the BRDF model at the Gaussian's center as done by [10] is too sparse. Therefore, we propose a deferred shading approach for Gaussians, where we evaluate the BRDF model in image space after the rasterization of the Gaussians. The image space surface position is projected back to the 3D space to evaluate the BRDF model at this surface point. This way, high-frequency details such as specular highlights are properly reproduced. In addition, a large range of editing capabilities is enabled as pixel operations can be applied to the shading buffers.

With this formulation, we achieve joint learning of geometry and the direct and global appearance of SSS objects. See Appendix F for more details on the architecture and the training. Further, the choice of the 3D GS representation and our lightweight MLP combined with explicit PBR shading allow for real-time rendering speeds. Due to the deferred shading approach, we can capture high-frequency details such as specular highlights. Furthermore, the decomposition into explicit distinct appearance components allows for a high degree of editability, even controlling the degree of SSS effects.

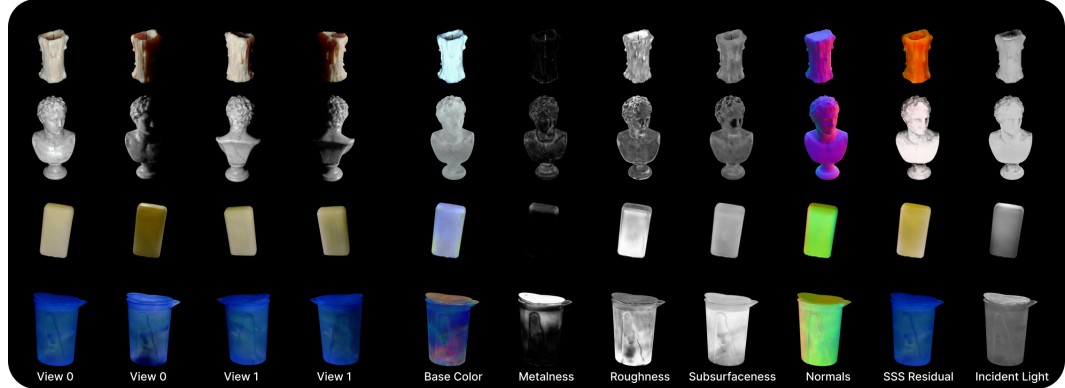

Figure 3: **Results of Decomposition** – showing two different views with different light directions. Further, the decomposition of PBR parameters is shown. The first two objects shown are synthetic while the lower two are scanned real-world world objects.

## 4 Experiments

Our proposed method facilitates real-time rendering of SSS objects and enables relighting and material editing. In the following, we present results of our method which is evaluated on synthetic and real-world datasets.

### 4.1 Experimental setup

**Dataset** While other NeRF-based SSS reconstruction and novel view relighting methods exist [46, 49] none of them provide a public dataset. We created a new OLAT dataset from synthetically rendered objects and real-world captured objects that capture various effects of SSS materials. In total, our datasets consist of 20 distinct objects from translucent material categories such as plastic, wax, marble, jade, and liquids (Sec. B).

For the **synthetic dataset** we created a synthetic light stage setup in Blender [8] that follows the formulation of [49]. It models 112 fixed light positions on the upper hemisphere divided into 7 rings of 16 lights each. For training, we render 100 random camera views of each object with a fixed distance to the object. For testing, we use the NeRF synthetic camera path proposed by Mildenhall et al. [27], which consists of 200 camera views positioned outside the light stage hemisphere. In total,

we have $11.200$ train and $22.400$ test images of $800 \times 800$ resolution for each object. We rendered datasets for 5 distinct objects with Blender's [8] Cycles renderer and the Principled BSDF shader. The 3D models are sourced from BlenderKit library [2].

The **real-world dataset** was captured in a light stage with a turntable supporting the object and a camera mounted on a motorized sled which can move on the vertical main arc of a sphere. Currently, the dataset includes 15 objects selected for a diverse representation of geometric detail and materials exhibiting SSS based on local availability and suitability for the acquisition setup in terms of rigidness and size. We captured $158$ positions per object with $167$ light positions each. Additionally, one image with uniform illumination was captured for camera pose optimization and object masking. The relative camera positions were optimized with COLMAP [34] resulting in an inward-facing $360°$ multi-view dataset. The cameras were aligned to the known light source positions in a joined reference frame. Similar to the camera reconstruction, object masks are generated from the uniformly lit images using automatic image matting based on [42]. Depending on the object we use a text prompt or points from the SfM stage as query to first generate pseudo trimaps leveraging SAM [21] and an open-vocabulary model [24]. The matting is then performed by ViTMatte [43] yielding masks with transparency. See appendix section A for more information on the image processing pipeline. We will release more details when publishing the dataset.

A total of approximately 25,000 images are split evenly into a train and test set by uniform sampling from the camera and light positions. We exclude frames with strong light flares or incomplete or wrong masks based on heuristics and some manual annotation. In summary, we discard roughly 10 % of the dataset. We don't always use the full size of the datasets for training, as shown in Table 1. Our method can also be trained sparsely using only $500$ images.

**Implementation Details**   SSS GS builds upon the 3D GS framework [18] and its extension Relightable 3D Gaussians (R3DGS) [10] that we in turn extend to capture the subsurface scattering effects of an object. For our implicit representation of the scattering component and the incident light prediction, we use a shallow MLP with 3 layers with Leaky-ReLU activations [11]. The whole pipeline is implemented with Pytorch using the provided custom CUDA kernels from [19, 10] for rendering. For more details on the training setup see Appendix F.

## 4.2   Qualitative Results

Our qualitative results are shown in Figure 3 and Figure 6. However, relighting and novel view synthesis are best experienced in the supplementary video. In 3 we show results rendered from our test set of objects from the synthetic (first two rows) and real-world (bottom two rows) part of our newly created dataset. In addition to novel view synthesis (view 0, view 1) we can freely change the light direction which we show for both views. We present the decomposition of the object into the surface appearance parameters, SSS effects and incident light map as they are used to render the final views on the left. Our method captures the SSS effect well using the SSS residual and adds the object's specularity from the PBR shading. Together with the predicted incident light the model achieves a plausible decomposition into basecolor, roughness and metalness material parameters. Note how the volumetric component can represent the complex light transport inside the entire volume. This is particularly observable in the case of the Tupperware object while the normals show a detailed representation of the surface. Find further quantitative analysis regarding the decomposition in section E.

**Comparison**   In Figure 5 we show that Relightable 3D Gaussian (R3DGS) [10] fail to capture the subsurface scattering effects of the object. As they only allow training on a static scene without any dynamic lighting the scattering component is baked into the basecolor which will fail to represent the SSS for a new light position (see also Sec. C) Our method can capture the subsurface scattering effects and the specular highlights of the object. Achieving clearer results than R3DGS [10] in that regard due to the formulation of shading in image space. Figure 5 also visualizes the difference between shading in world space vs. our deferred shading approach in image space. As visualized on the left the Gaussians occupy different, sometimes very large areas along the surface limiting the rendering of high-frequency effects like specular highlights. NeRF-based methods have a lot more training capacity due to their large MLPs that can represent the complex light transport connected to SSS. For a fair comparison we select KiloOSF a variant of OSF [46] that is optimized for real-time rendering. Compared to KiloOSF [46] in Figure 6 it becomes apparent that also a small MLP paired

with our explicit shading approach can achieve higher quality results after a shorter training time. Even after 20 hours of training, there are still some artifacts visible in the geometry that stem from the voxelization performed by the underlying KiloNeRF [31]. While the Stanford Bunny object overall works well, the soap bar and car toy object are lacking in the representation of the surface reflectivity and show frequent errors in the geometry.

## 4.3 Quantitative Results

We evaluate our method on the test split of the synthetic SSS dataset and the real-world SSS dataset on the task of novel view synthesis using the following metrics: PSNR, SSIM, and LPIPS. The results are shown in Table 1. Our method can successfully render novel view synthesis and relighting of subsurface scattering objects at real-time speeds, with high-quality results. Each component of our method is crucial to achieving such results, we show the ablation of the components in the appendix (Sec. C).

| Method | Category | PSNR↑ | SSIM↑ | LPIPS↓ | FPS | Train T. | Res. | Data |
|--------|----------|-------|-------|--------|-----|----------|------|------|
| **Ours** | Synthetic | $37.35 \pm 2.13$ | $0.986 \pm 0.006$ | $0.03 \pm 0.01$ | $161.2 \pm 11.95$ | $\approx$ 2h | $800^2$ | 11.200 |
| **Ours** | Real World | $31.12 \pm 2.11$ | $0.96 \pm 0.02$ | $0.042 \pm 0.03$ | $155.25 \pm 22.38$ | $\approx$ 2h | $800^*$ | $\approx 12.000$ |
| **Ours** | Synthetic | $\mathbf{35.01} \pm 1.01$ | $\mathbf{0.972} \pm 0.01$ | $\mathbf{0.040} \pm 0.01$ | $\mathbf{154.8} \pm 28.26$ | $<$ 1h | $256^2$ | 500 |
| KiloOSF | Synthetic | $25.91 \pm 1.88$ | $0.93 \pm 0.02$ | $0.097 \pm 0.03$ | 14.4 | $>$ 20h | $256^2$ | 500 |
| **Ours** | Real World | $\mathbf{26.61} \pm 0.09$ | $\mathbf{0.93} \pm 0.003$ | $\mathbf{0.08} \pm 7\text{e-}4$ | $\mathbf{94.5} \pm 3.54$ | $<$ 1h | $256^2$ | 500 |
| KiloOSF | Real World | $23.24 \pm 1.58$ | $0.83 \pm 0.09$ | $0.21 \pm 0.07$ | 14.4 | $>$ 20h | $256^2$ | 500 |

Table 1: Quantitative results on test views of our mehod on large images and a bigger dataset (top) and comparison against KiloOSF [46] within their setting (bottom). We excluded runs of KiloOSF that couldn't reconstruct anything for more comparable results. The best results are highlighted in bold. All experiments where timed and run on a single NVIDIA RTX 4090. *: cropped images with one side length of 800.

**Comparison** We compare our method against the state-of-the-art NeRF method KiloOSF [46] which claims to achieve rendering at real-time speeds ($<$ 14 FPS). We use a single NVIDIA RTX 4090 GPU per run on a compute server with a total of 512 GB of RAM. We use the provided code and default configuration for the experiments, converting our datasets to their dataset specification. To be comparable and to fit within the framework of [46] we downscale our images to a resolution of $256 \times 256$ and only use 500 images of our synthetic dataset. We want to point out, that our method achieves similar results at real-time speeds on images of $800 \times 800$ resolution. In comparison, our method is faster to train since we can carry over the efficiency of the 3D GS framework by carefully designing our pipeline around a small MLP and an explicit BRDF decomposition. Similarly, we can save on the number of parameters per Gaussian since we directly store color instead of the SH coefficients, for example.

Up until now we only compare within the domain of real-time rendering and novel view synthesis, for SSS reconstruction and relighting, as our method is optimized for speed and editability through decomposition. Comparing the numbers of Tabel 1 with the results reported in subsurface NeRF-based methods that aim for quality [49, 25] we still achieve similar results within the domain of synthetic scenes, and are en-par on real-world scenes. However, the datasets on which they reported the measurements are not publicly available. Therefore, we cannot directly compare.

## 4.4 Applications

In Figure 4, we provide a detailed demonstration of the editing applications enabled by our approach. Our method facilitates adjustments to the base color and SSS residual, allowing for seamless color changes. Additionally, it supports the editing of material parameters, enabling modifications that make the appearance more metallic, shinier, or rougher. We also demonstrate the ability to alter the opacity and intensity of the SSS residual, and show single light illumination beyond the training domain. Moreover, our approach introduces editing capabilities, powered by the SSS residual, that surpass those available in previous methods [10].

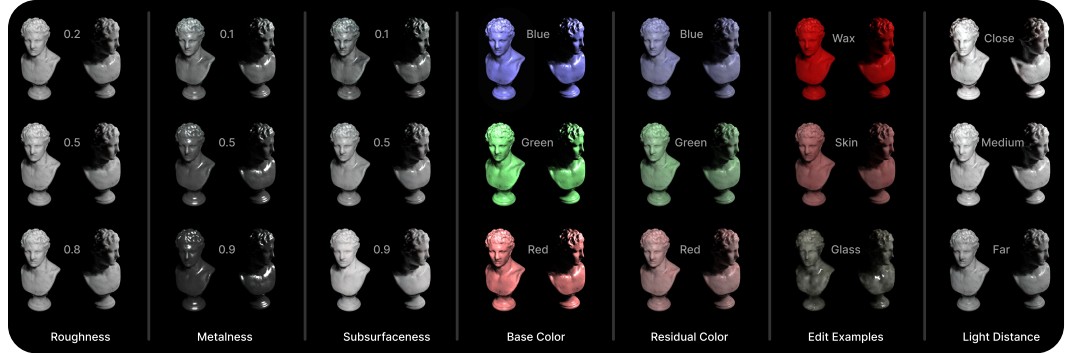

Figure 4: **Editing** results, showcasing PBR based edits such as (roughness / metalness / base color) as well as method specific properties (subsurfaceness / residual color). The latter highlights editing only possible with this method. The rightmost column shows light positions not sampled from the light stage.

Find details on further qualitative and quantitative analysis of the application possibilities of our method such as image based relighting (Sec. D) in the appendix.

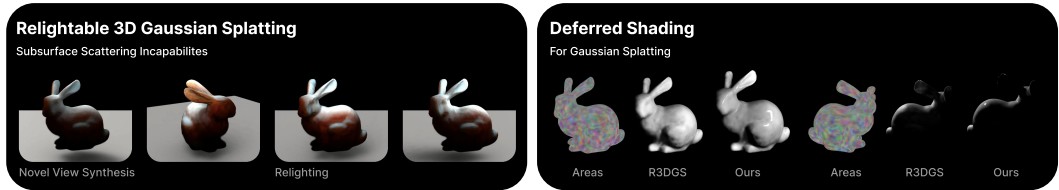

Figure 5: **Limits of Relightable 3D Gaussians (left)** – While Relightable 3D Gaussians can reproduce view and illumination-dependent reflections they fail to properly relight subsurface scattering objects.
**Deferred Shading (right)** – allows us to evaluate the surface reflectance for each rendered pixel instead of per Gaussian. This way, specular highlights are rendered with crisper detail.

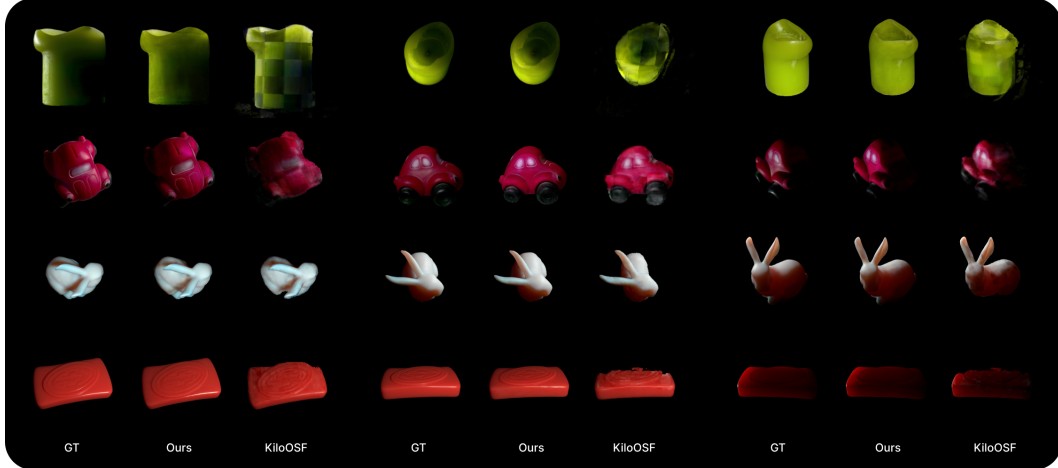

Figure 6: **Comparison** against the KiloOSF [46] method. The top two objects are real-world objects and the bottom two synthetic objects. Note the qualitative improvement in shape and appearance compared to KiloOSF.

## 5   Limitations

Our approach reproduces SSS by modeling the apparent effect at the Gaussian where the light leaves the object. To avoid the costly integration over the whole object surface, this information is predicted by an MLP. Intricate variation due to strongly heterogeneous materials or angularly dependent SSS effects might only be roughly approximated. Furthermore, altering the geometry or the BSSRDF of the object would require retraining the SSS representation. While the screen space shading improves specular rendering significantly the shadowing is still limited by the low resolution

of the 3D Gaussians in the currently used rendering scheme. Additionally, our screen space shading complicates the modeling of transparent refracting surfaces as it would require multiple passes to correctly warp the buffers. We do not explicitly account for this in our method. The white light assumption helps the severely under-constrained decomposition task. Still, while the decomposition is constrained to be physically plausible, it might not always match the ground truth as there are a multitude of possible explanations given only the appearance and light position. Moving forward we would like to also enable reconstruction under a less constrained illumination setting. Although not the intended use case and scope of this work our method could potentially be used to improve the scanning of humans as SSS is an integral property of the appearance of human skin. This has implications on personal rights and privacy and potential misuse that need to be addressed in these cases.

## 6    Conclusion

We present Subsurface Scattering (SSS) for 3D Gaussian Splatting (3D GS), a method to reconstruct translucent objects from OLAT multi-view image sets. By decomposing light transport into explicit PBR materials and an implicitly represented scattering component we enable novel view synthesis and relighting, as well as light and material editing in real-time. A per 3D Gaussian SSS parameter is learned to merge the two components. Our formulation enables a small MLP to reason about local and global light transport in the scene and predict incident light in addition to the SSS radiance. By evaluating the BRDF in image space we can achieve high-quality specular shading independent of the resolution of the 3D Gaussians. Compared to 3D GS we enable high-quality reconstructions for a new class of objects and achieve faster optimization and rendering speed than previous NeRF-based methods with similar or improved quality. Some limitations connected to the SSS representation and the rendering scheme remain which open up an interesting trajectory for future work. We also plan to release our dataset as the first OLAT SSS dataset including real-world translucent objects to the community.

## Acknowledgements

Funded by the Deutsche Forschungsgemeinschaft (DFG, German Research Foundation) under Germany's Excellence Strategy – EXC number 2064/1 – Project number 390727645. This work was supported by the German Research Foundation (DFG): SFB 1233, Robust Vision: Inference Principles and Neural Mechanisms, TP 02, project number: 276693517. This work was supported by the Tübingen AI Center. The authors thank the International Max Planck Research School for Intelligent Systems (IMPRS-IS) for supporting Jan-Niklas Dihlmann and Arjun Majumdar. In cooperation with Sony Semiconductor Solutions -Europe, the authors would like to thank Mr. Zoltan Facius and Mr. Prasan Ashok Shedligeri.

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

# A   Appendix

## A   Camera Preprocessing Pipeline

For capturing the real-world datasets (Fig. 7) we use an Oryx 123S6C-C camera by Teledyne/FLIR. The camera has a resolution of $4092 \times 3000$ pixels and uses a Bayer pattern to capture color. The white balance settings for the camera and light sources were estimated using an X-Rite Color Checker. We capture raw, single-channel 8-bit images to render the data transfer from the camera as fast as possible. To reduce noise we always capture 5 frames instead of one and compute the pixel-wise median. We remove fix-pattern noise via dark-frame subtraction. We capture a median-filtered dark frame for every object in the dataset. The median-filtered and dark frame corrected images are demosaiced using OpenCV [5]. For every view direction, we capture one image with all light sources in the Light Stage active. We run a structure from motion (SfM) algorithm from COLMAP [34] on those uniformly lit images to get the precise camera positions. The objects were placed on a box with a random noise texture to aid feature detection.

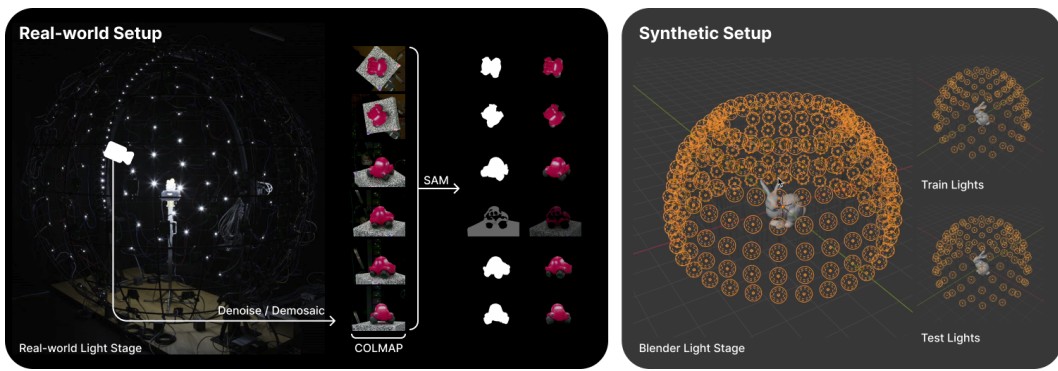

Figure 7: **Preprocessing and Light Stage** showing our real-world and synthetic data acquisition pipeline. For the real-world data, we present a sample of the all lights on images used for COLMAP [34] reconstruction. Further we show our segment anything [21] based automatic masking of the data (with a failure case that we filter out for training). Note that all lights on images are not used during the training process. On the right, we show the synthetic data acquisition pipeline and the train and test split also applied to the real-world data.

## B   Translucent Object Dataset

Our datasets consist of 20 distinct objects from translucent material categories such as plastic, wax, marble, jade, and liquids. We captured 15 object in total with our light stage setup and processed them as described in Section A. Further, we constructed 5 synthetic scenes using Blender [8] and rendered them in a synthetic light stage setup. Figure 8 shows our entire dataset. Find the full dataset on our project page.

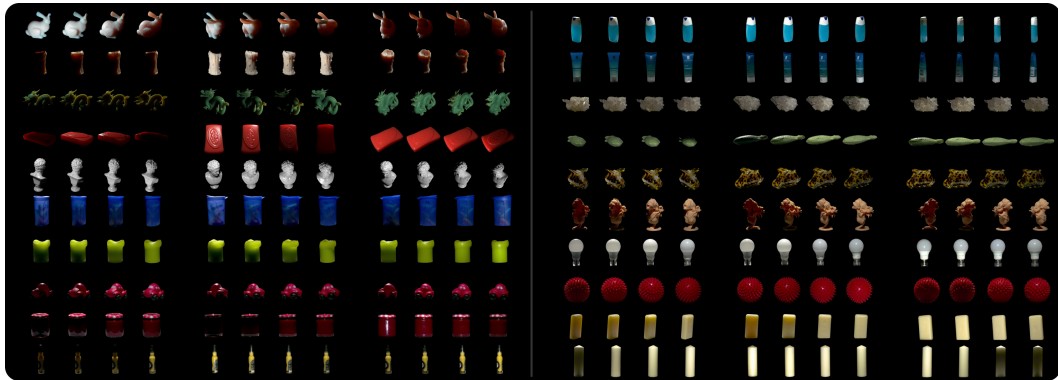

Figure 8: **Translucent Captured Objects** highlighting a subset of images from various camera and light poses with all 20 objects of our dataset. The top left 5 are synthetic and the remainder shows real-world captures.

## C   Ablation

We conducted an ablation study to evaluate the importance of each component of our method (Tbl. 2). The first variant of our method removes the SSS residual and solely relies on our PBR shading model. As clearly visible (Tbl. 2) results degrade significantly, highlighting the impracticability of solely relying on the PBR shading model for translucent objects. We also compare the results of our full method against a variant without the PBR shading model and a variant without the deferred shading stage. The results show that the PBR shading stage together with deferred shading is crucial for the generation of high-quality images, as it enables the representation of specular highlights. Please note that the metrics used only have limited capability to express the fine details in visual quality that the improved representation of specular highlights adds (Fig. 9).

| Method | Category | PSNR↑ | SSIM↑ | LPIPS↓ |
|---|---|---|---|---|
| Ours | Full | **36.16** ± 3.76 | **0.982** ± 0.010 | **0.027** ± 0.006 |
| Ours | w/o Deferred | 35.61 ± 2.28 | 0.981 ± 0.009 | 0.027 ± 0.004 |
| Ours | w/o PBR | 35.00 ± 1.98 | 0.981 ± 0.009 | 0.029 ± 0.005 |
| Ours | w/o Joint MLP | 31.68 ± 0.66 | 0.971 ± 0.002 | 0.041 ± 0.008 |
| Ours | w/o Residual | 30.23 ± 2.28 | 0.954 ± 0.015 | 0.056 ± 0.026 |
| R3DGS [10] | w/ Inc. Light F. | 24.81 ± 5.29 | 0.926 ± 0.055 | 0.087 ± 0.044 |

Table 2: **Component Ablation** we show ablations on a subset of the dataset consisting of two synthetic and two real world scenes. "Full" refers to the method used in the paper. We trained four variants of our method without crucial components "w/o Deferred", "w/o PBR", "w/o Joint MLP" and "w/o Residual". The last row adds a comparison of a variant of R3DGS [10] which combines R3DGS with our incident light field.

Further ablations highlight the importance of the joint MLP. If we split the MLP for residual and incident light, inputting only the relevant physical properties, it leads to worse results. We believe this is because, with the joint parameterization, the main MLP can learn about the global light transport, thereby providing valuable insights for the two output heads predicting the SSS residual and the incident illumination, respectively.

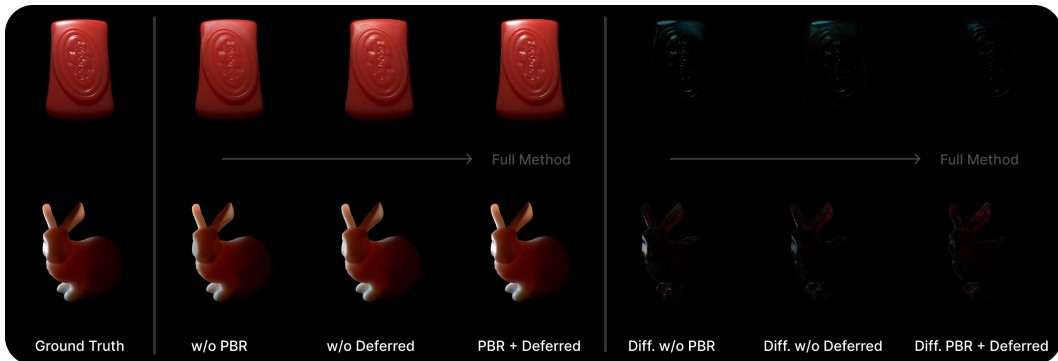

Figure 9: **Shading Ablation** highlighting the importance of each component of our method and their difference compared against ground truth. First (left to right) "w/o PBR" (only residual and incident light field), second adding PBR in "w/o Deferred" and last showcasing full specular highlights and the least error by employing our full method "PBR+Deffered". We show results from individual training runs (not post-training toggling of components).

We also run experiments against a variant of R3DGS [10] with known illumination. Please note that the original R3DGS approach only supports a single illumination setting, therefore differing from our OLAT setting and not capable of learning to disentangle SSS and surface reflection. To still work within the OLAT setting we attached the incident light field prediction from our method to the R3DGS pipeline. The results in Table 2 indicate the limitations of the base model in our experiment setting. In summary, the ablation study demonstrates the importance of each component of our method and the necessity of the joint MLP for the prediction of the SSS residual and incident light field.

## D Image Based Lighting

In Figure 10 we show that our method achieves image based lighting (IBL) with high visual quality. First, we sample the HDR maps representing distant environment illumination. The samples don't need to correspond to OLAT samples used in training and could be placed much denser. Using this approach we can generate a relit frame in about 20 seconds for a medium resolution. To speed up generation we can precompute a reflectance field assuming white light. We compute the relit view as the sum over the reflectance field scaled by the environment illumination before applying tone mapping for display. This runs in a fraction of a second even with a naive implementation. As can be seen in Figure 10 an object can be rendered in different illumination settings [14] yielding consistent photorealistic results.

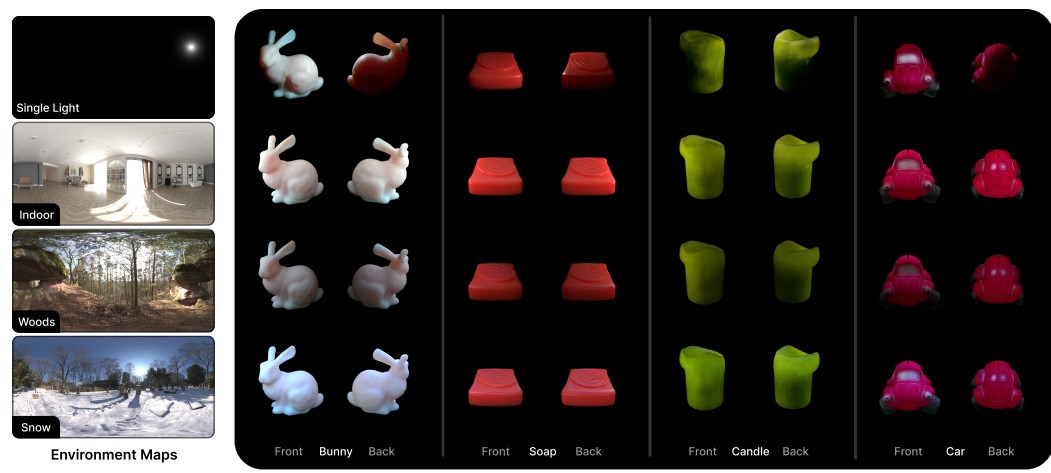

Figure 10: **Image Based Lighting** (IBL) results of our method, with an OLAT sample in the top row for comparison and three different environment maps used on two synthetic (bunny / soap) and two real-world (candle / car) objects.

We also compared quantitatively against Blender [8] renders for chosen environment maps. For fairness, we used the same sampling of the environment map as point lights as in our method. As can be seen in Table 3, the relighting results closely resemble the ground truth. Note, that the PSNR metric is affected by denoising artifacts from Blender and the noise residual from the Monte-Carlo path tracing in the ground truth data.

| Environment Map | PSNR↑ | SSIM↑ | LPIPS↓ |
|---|---|---|---|
| Average | **31.62** ± 2.31 | **0.974** ± 0.012 | **0.049** ± 0.018 |
| Indoor | 30.24 ± 1.06 | 0.971 ± 0.0133 | 0.052 ± 0.019 |
| Woods | 34.36 ± 1.37 | 0.978 ± 0.009 | 0.044 ± 0.017 |
| Snow | 30.28 ± 1.31 | 0.971 ± 0.013 | 0.050 ± 0.019 |

Table 3: **IBL Quantitative** comparison of relighting our five synthetic scenes against Blender [8] renders using environment maps from Figure 10 above.

## E Intrinsics

We further evaluated the intrinsic properties (Fig. 11) referring to albedo and illumination as common properties of intrinsic image decomposition. While our base color parameter can be understood as the albedo we also output additional material parameters that are needed for our physically based rendering model. We show base color, roughness, metalness, normal, sss residual, specular & diffuse components and again the final render. Specular and diffuse illumination are intermediate results in our rendering pipeline during the deferred shading stage representing the diffuse and specular illumination components (before multiplication with the base color), respectively.

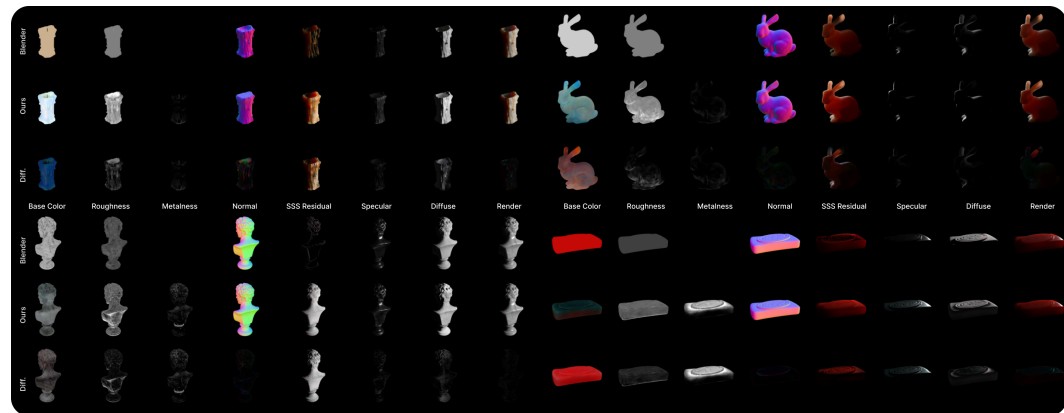

Figure 11: **Intrinsics Maps** rendered from Blender compared to ours with absolute difference at the bottom. Blender does not provide SSS intrinsics. Therefore, the residual here is the difference of the diffuse light rendered with SSS on and off. Find in Tab. 4 results for all synthetic scenes.

| Type | Base Color | Roughness | Metalness | Normal | Residual | Specular | Diffuse | Render |
|------|-----------|-----------|-----------|--------|----------|----------|---------|--------|
| Bunny | 0.152 | 0.037 | 0.019 | 0.018 | 0.018 | 0.017 | 0.163 | 0.057 |
| Dragon | 0.077 | 0.141 | 0.045 | 0.366 | 0.039 | 0.022 | 0.074 | 0.190 |
| Statue | 0.119 | 0.038 | 0.026 | 0.049 | 0.016 | 0.013 | 0.159 | 0.065 |
| Soap | 0.164 | 0.047 | 0.031 | 0.203 | 0.012 | 0.013 | 0.081 | 0.061 |
| Candle | 0.080 | 0.042 | 0.014 | 0.015 | 0.032 | 0.013 | 0.090 | 0.065 |
| Average | 0.119 | 0.061 | 0.027 | 0.130 | 0.023 | 0.016 | 0.113 | 0.087 |

Table 4: **Intrinsic Comparison** of Blender renders against ours. We report RMSE for all synthetic measurable scenes, a subset of 25 renders with 5 different camera and light poses, and "Average" over all of our synthetic scenes.

All selected properties are compared to ground truth properties obtained using the Cycles renderer in Blender [8] for our synthetic scenes. We want to note that Blender uses a different shading model, such that some of these properties are not directly equivalent. Most notably, the SSS residual cannot be retrieved and is calculated by us as the difference of diffuse reflection from a rendering with SSS turned on and SSS turned off. In Figure 11 we plotted the absolute difference and properties. For all of these intrinsics, we calculated the RMSE, see Table 4. While achieving overall good results, especially for the illumination, the value of the quantitative evaluation is limited by the fact that the optimization can generate multiple plausible solutions for a given appearance due to the under-constrained problem space. This is also the reason why base colors might differ.

## F Training Details

In this section, we provide details on the training of our model, including the loss terms, network architecture, and optimization details. For training, we use the PyTorch framework and train on a single NVIDIA RTX 4090 GPU with 24GB of memory. Our code is build upon the 3D Gaussian Splatting (3D GS) [18] and Relightable 3D Gaussians (R3DGS) [10] codebase. For further details on the training pipeline, we refer to the original papers and our codebase, which you can access via our project page.

**Losses** We have multiple loss terms in our training pipeline that are mainly adapted from R3DGS [10] that we will briefly outline them and their weighting here. As in 3D GS [18], we utilize a $L_1$ loss and perceptual loss $L_{\text{SSIM}}$ comparing the predicted and ground truth images. Both of those are combined with $L_{\text{img}} = (1.0 - \lambda_{\text{dssim}})L_1 + \lambda_{\text{dssim}}(1.0 - L_{\text{SSIM}})$ where $\lambda_{\text{dssim}} = 0.2$. We further optimize for the perceptual quality of the images by employing a LPIPS loss that is weighted with $\lambda_{\text{lpips}} = 0.2$. Taken from R3DGS [10] we utilize a MSE between pseudo normals calculated from the depth map and the predicted normals, scaled with $\lambda_{\text{normals}} = 0.02$. Special to our work we constrain the clamped predicted incident light to be close to the evaluated visibility with an $L_1$ loss and $\lambda_{\text{incident}} = 0.02$. This helps to avoid evaluating to zero in rare cases and thus only utilizing the MLP for the prediction. As in R3DGS we penalize the creation of Gaussians outside the masked original image

space by using a mean entropy loss $L_{\text{mask}} = -(I_{\text{mask}} \log(\text{opacity}) + (1 - I_{\text{mask}}) \log(1 - \text{opacity}))$ with $\lambda_{\text{mask}} = 0.1$. We further employ R3DGS bilateral smoothing losses for the predicted metalness, roughness, subsurfaceness and base color with $\lambda_{\text{smooth}} = [0.002, 0.002, 0.002, 0.006]$. Additionally, we use the highlight and shadow enhancement loss from R3DGS with $\lambda_{\text{enhance}} = 0.005$ and the raytraced visibility learning that is weighted with $\lambda_{\text{raytrace}} = 0.01$.

**Network** As outlined in Figure 2 we use a shallow MLP to predict the subsurface properties and incident light of the Gaussians. Our MLP accepts an input having $(16 + 24)$ feature dimensions, with normals (3), rotation (2), scale (3), light direction (3), view direction (3), light distance (1) and visibility (1). The position (3) is encoded using a positional Fourrier encoding [27] resulting in 24 features. The input is fed into a 3-layer MLP with $[64, 32, 32]$ neurons in them followed by Leaky-ReLU activation functions. The output is fed into two output heads. One of them predicts the incident illumination with another layer of 32 neurons and a Leaky-ReLU and a ReLU output layer, while the other one computes the SSS component of the radiance with a sigmoid output layer.

**Optimization** The per-Gaussian position $\mu$, covariance as rotation $q$ and scale $s$, opacity $o$, basecolor $c_{base}$, metalness $m$, roughness $r$, normal $n$, light visibility $v$ as Spherical Harmonics and the newly introduced subsurfaceness $sss$ are optimized together with the network weights for the base MLP and the two output heads for SSS radiance and incident radiance, respectively. We use the ADAM [20] optimizer with default parameters and a learning rate of $0.001$ with an exponential decay of $0.99$ every 1000 steps. We train for $60k$ steps although we observe that the model already receives good results after $30k$ steps. Further, we follow the default splitting and pruning schedule proposed by the original 3D GS.

**Scheduling** We schedule the training of the individual 3D Gaussian properties with the original provided exponential learning rate decay function. For our MLP we also use an exponential decay function with a gamma of $0.9999$. Further, we schedule the incident light by linearly incrementing the incident light up until 7k steps and then removing the constraint. This helps to stabilize the training and prevent the MLP from predicting zero incident light in the beginning. We also freeze the optimization of the roughness and set it to a value of $0.5$ for the first 10k steps to stabilize the training.

