# OpenReview forum: "Subsurface Scattering for Gaussian Splatting"
_NeurIPS.cc/2024/Conference — NeurIPS 2024 poster_

### Official Review · Reviewer_uk4p · 2024-06-23

**Soundness:** 3
**Presentation:** 3
**Contribution:** 2
**Rating:** 6
**Confidence:** 4

**Summary:**

This paper proposes a framework for capturing the geometry, specular, and subsurface scattering appearance of 3D objects using a captured dataset composed of multi-view OLAT images.
The appearance of the object is decomposed into two different models.
First, a 3D Gaussian representation with a spatially varying BRDF model for explicit surface representation. Secondly, an implicit volumetric representation is used for subsurface scattering appearance.
This framework enables material editing, relighting, and novel view synthesis in real-time.

**Strengths:**

1. I believe that the authors will provide a multi-view OLAT dataset for subsurface scattering objects in the future, and this dataset will be a great contribution to the vision and graphics community.
2. Screen space shading algorithm for 3D Gaussian representation improves the view-dependent appearance quality of the reconstructed object.
3. The use of two different appearance models enables the reconstruction of the geometry, BRDF, and subsurface scattering of 3D objects.

**Weaknesses:**

1. Missing references for diffusion-based SSS approximation models:

[1] A Practical Model for Subsurface Light Transport, Jensen et al., 2001.

[2] Light Diffusion in Multi-Layered Translucent Materials, Donner et al., 2005.

2. Lack of validation on the intrinsic properties.

To generate realistic novel relit scenes and material edits, it is important to correctly acquire the intrinsic properties. For the synthetic data, each intrinsic property can be directly compared with the reconstructed one (RMSE). Although reconstructed properties can differ due to the different models from Blender or the limitations of the method, the author should address this. Currently, it is unclear if this method reconstructs each intrinsic property well because there is no ground truth intrinsic property information in the paper or supplemental material, even though the rendered results look realistic and similar to the ground truth data. For example, in the dragon dataset in the supplemental material, there are many specularity changes in the ground truth, but these are not observed in the renders.

3. The relighting results only include novel views with single-light images.

There are no results for environmental lighting or changing light colors. Since the framework does not support multiple light sources, we could generate multiple light sources or environment-relighting results similar to the previous method [3]. Simply adding multiple OLAT images in screen space would provide much more powerful relighting results, which I would like to see.

[3] Neural Light Transport for Relighting and View Synthesis, Zhang et al., 2020.

4. It would be powerful to show the change in intrinsic properties between objects, for example, from bunny to dragon and vice versa.

**Questions:**

1. The real-world dataset images look over-exposed in the specular regions. In the experimental setup. In the paper, same images were captured five times to reduce image noise. Did you change the exposure of the camera while capturing the raw images? Moreover, why did you choose 8-bit instead of 16-bit images?
2. This doesn't need to be addressed in the rebuttal (Minor), but it would be better to add some information about the name of each video. Without clear explanations, some notations can be misleading to the reader.

**Limitations:**

The author adequately addressed the limitations, including:
1. Difficulty in optimizing strongly heterogeneous materials.
2. Limitations in screen space shading.
I might add that this paper is also limited to static scenes only. Human skin, for example, is constrained not only by personal rights but also by the inability to handle movement during capturing. This will cause blurry results that the current framework cannot handle.

---

> ### Author Rebuttal · Authors · 2024-08-06
>
> ### Additional Relighting Results with Image Based Lighting
> > Weakness 3
> Please find qualitative and quantitative results for the image based lighting in Fig. 1 and Tab. 1 in the PDF accompanying this rebuttal. We also show additional relighting results together with the other editing capabilities in Fig. 2. Using our approach as outlined in the global response in  “Image Based Relighting” we can generate a relit frame in a fraction of a second after a one-time pre-computation of the reflectance field which takes 20 seconds for a reasonable resolution. Alternatively, we can perform importance sampling on a given environment map which reduces combined processing time but cannot be reused for a different illumination setting.
>
> ### Evaluation of Intrinsic Properties
> > Weakness 2
> We provide a small qualitative evaluation in Fig. 4 and an extensive quantitative evaluation in Tab. 3 on the prediction of intrinsic properties of our model.
> We assume the reviewer refers to albedo and illumination as common properties of intrinsic image decomposition. While our base color parameter can be understood as the albedo we also output additional material parameters that are needed for our physically based rendering model. We think it is interesting to also compare these properties. Therefore, we decided to showcase base color, roughness, metalness, normal, sss residual, specular & diffuse component and again the final render. Specular and diffuse illumination are intermediate results in our rendering pipeline during the deferred shading stage representing the diffuse and specular illumination components (before multiplication with the base color), respectively.
>
> > All selected properties are compared to ground truth properties obtained using the Cycles renderer in Blender [8] for our synthetic scenes. We want to note that Blender uses a different shading model, such that some of these properties are not directly equivalent. Most notably, the SSS Residual cannot be retrieved and is calculated by us as the difference of diffuse reflection from a rendering with SSS turned on and SSS turned off. In Fig. 4 we plotted the absolute difference and properties for one example and will add more to the supplementary material. For all of these intrinsics we calculated the RMSE. While achieving overall good results, especially for the illumination, the value of the quantitative evaluation is limited by the fact that the optimization can generate multiple plausible solutions for a given appearance due to the under-constrained problem space. This is also the reason why base colors might differ.
> As correctly noticed by the reviewer, the dragon scene does not capture all specularity changes and is also the one performing the worst in our analysis. It is the most complex scene with heterogeneous subsurface material and a lot of geometric details. Our method does not perfectly represent the geometric detail of the dragon scales in the scene resulting in incorrect specular highlights that lead the optimization to a rougher material. To improve our method for such tough scenes and also to better handle anisotropic scattering are things we want to explore in future work.
>
> ### Editing of Intrinsic Properties
> > Weakness 4
> For the editing of intrinsic properties we want to reference Fig. 2 that highlights free editing of intrinsics and shows results for various settings of those properties. Surely, intrinsic properties could be transferred between models also shown by edit examples in Fig. 2 transforming the statue into various different material types. Note, that some intrinsics such as the residual are position dependent and could hardly be transferred, though.
>
> ### Multi-exposure Acquisition
> > Question 1
> > We acknowledge HDR images in the dataset could be beneficial, especially for fitting BRDFs. We still decided to capture our real-world dataset in LDR for three main reasons, which all boil down to ease of use:
> > - In deep learning working with LDR images is much simpler than handling the unconstrained value ranges of linear HDR images.
> > - Competing methods are also designed for and trained on LDR datasets.
> > - The disk space needed for capturing, processing and later hosting and downloading such a big HDR dataset would be substantially higher than for a compressed LDR dataset. The png-compressed, single channel, 8-Bit Bayer images still need 2.2 TB of storage, which increases to 5.2TB when adding color, rectification and object masks.
> >
> > While some specular highlights will be unavoidably clipped the overall exposure is carefully set for an optimal SNR for single light images as can also be seen in Fig. 6 of the rebuttal PDF.
>
> ### Additional References on Diffusion Based SSS Estimation
> >  Weakness 1
> We reworked the related works section also including the suggested literature on diffusion based estimation of scattering parameters.
>
> ### Presentation and Labeling of Videos
> > Question 2
> We thank the reviewer for the additional feedback regarding the presentation of the results. We will take this into account when redesigning the supplementary material and accompanying website.
>
>
> > We want to thank the reviewer again for the thorough analysis of our work. We hope that our new evaluation of intrinsic properties as well as the relighting results add insights into the method’s capabilities and that we could clarify the reasoning for some of our choices for the data acquisition.

---

> ### Comment · Reviewer_uk4p · 2024-08-13
>
> I really appreciate the results related to the validation of the intrinsic properties, editing materials, and the novel environment rendering which were pointed out as a major weakness in this review process have been appropriately addressed. I also fully understand the issues related to the dataset size. I believe that if the paper is reorganized to include the rebuttal content, it would be sufficient for acceptance. However, significant level of revisions would be necessary due to the amount of rebuttal. I changed the rating to weak acceptance for now, but I’m leaning more towards borderline acceptance or low. I will take into account the opinions of the other reviewers as much as possible.

---

> > ### Author Response · Authors · 2024-08-13
> >
> > We sincerely thank the reviewer for appreciating the additional results and the efforts we made to address the identified weaknesses. We have already incorporated the suggestions from the rebuttal into the paper, leading to an improved version.

---

### Official Review · Reviewer_uYWZ · 2024-07-08

**Soundness:** 2
**Presentation:** 3
**Contribution:** 2
**Rating:** 5
**Confidence:** 4

**Summary:**

This paper aims to model the subsurface scattering (SSS) effects under the 3D Gaussian Splatting framework, which is an efficient 3D representation for novel view synthesis. The challenge of this SSS modeling is the complicated light path in the final rendering output. The proposed framework is based on Relightable 3DGS but proposes an MLP-based module to capture SSS effects for 3DGS. It improves the representation of specularity by performing shading in image space. The experimental results achieve comparable or better results and fast optimization and rendering on synthetic and real-world data compared with NeRF-based approaches.

**Strengths:**

1. Introduce SSS residual to the 3DGS pipeline, which achieves comparable results and fast rendering speed.
2. Propose an image-space deferred rendering to capture specularity.

**Weaknesses:**

1. Some contributions may be overclaimed:
a. Editability. It was not demonstrated in the paper. Moreover, some editability comes with the Relightable 3DG, thus not the contribution of this paper.
b. The relighting effects seem only to change w.r.t the light direction.
c. Shadowing effects are handled by Relightable 3DG, thus not the contribution of this paper.
d. It is unclear how to make applications to other fields as mentioned in line 60, since this paper makes various assumptions like a single light source and multi-view captures.
2. Insufficient evaluation. As the method is based on Relightable 3DGS, it would be better to compare the proposed framework with Relightable 3DGS and 3DGS in quantitative results on novel view synthesis in Table 1.
3. The experiment settings are constrained: a single, known light source, object-centric, and given object masks. Moreover, the backbone model Relightable 3DG also requires estimated depth, which is not discussed in the paper.
4. Some typos and unclear sentences: line 232 about the number of images; Table 1 caption "non quadratic image". Line 298: sentence fragment. line 22 unclear sentence, etc.

**Questions:**

1. What is the performance of Relightable 3DG / 3DGS?
2. Eq. 6 overfits the experiment setting. Thus, it is hard to say the representation capabilities of the MLP. If the incident lighting in Eq. 6 is predicted and fixed by the MLP, how could the proposed method achieve relighting, like changing the intensity?

**Limitations:**

The authors have mentioned the limitations and potential negative societal impacts of their framework in Sec.5.

---

> ### Author Rebuttal · Authors · 2024-08-06
>
> ### Generalization of MLP / Representational Power of MLP
> > Question 2
> We add application examples that show interpolation capabilities in both the viewpoint as well as illumination domain. Results on the test set (Fig. 5 in the paper, for example)  are outside of  the training setting in terms of light and camera position. This can be best observed in the videos included in the supplementary material where also the light and camera distances are varied compared to the training.
> The global MLP is chosen specifically for its power to interpolate on the trained manifold similar to how NeRF [24] uses it.
>
> ### Additional Relighting Examples
> > Question 2, Weakness 1b
> Please find qualitative and quantitative results for the image based lighting application in Fig. 1 and Tab. . We also show additional relighting results together with the other editing capabilities in Fig. 2. As can be seen the object can be rendered in different, both indoor and outdoor illumination settings yielding consistent photorealistic results. Please also refer to the global response for additional notes on the relighting examples and the method of computing them.
>
> ### Editability
> > Weakness 1a
> We specifically focus on enabling intuitive and fast editing of the illumination and material properties of a reconstructed scene. In Fig. 2 we show examples of the material editing modes in detail. Fig. 1 shows new examples of relighting using environment lighting.
> As this topic was of interest for multiple reviewers we added a paragraph on “Editing Applications” in the global response that we invite the reviewer to take a look at, too.
>
>
> ### Limited Applicability to Other Domains due to Constraint Acquisition Setting
> > Weakness 1d, Weakness 3
> There are multiple application settings where a constrained acquisition setting like the OLAT approach might be tolerable and materials with subsurface scattering are relevant.
> In “Choice of OLAT Data and Relighting” of the global response we discuss the choice of the experiment setting. For example, in the games and movie industry a large effort is put into asset generation, often employing custom build scanning devices or light stages [https://home.otoy.com/capture/lightstage/] very similar to the one used in our dataset acquisition. Scans of individual objects that can then be composed and combined into larger scenes. Our relighting capability enables easy integration of objects into new settings. For scenes with multiple objects only the visibility attributes need to be updated as has been shown in [39].
> Also in the medical imaging domain it is feasible to perform a scan of e.g. an organ in a controlled environment to construct a detailed model that is then used for supervision during surgery, for example [47]. In this context also an endoscope with a point light source could be well represented using our model [48].
>
> ### Comparison against R3DGS and 3DGS
> > Question 1 & Weakness 2
> Additional results can be found in the newly added Tab. 2 as R3DGS (with incident light field). The results are obtained from a base configuration of our model without residual prediction and without deferred shading which comes closest to R3DGS.
> The original R3DGS approach only supports a single illumination setting which is fundamentally different from the OLAT setting with many different illumination configurations. See the paragraph on “Choice of OLAT Data and Relighting” in the global response for more information about the setting and its relevance to subsurface scattering estimation. We use a light representation that models the changing illumination with explicit point light locations compared to the NeILF [37] based approach in R3DGS. The results in Tab. 2 clearly indicate the limitations of the base model in our experiment setting.
> Plain 3D Gaussian Splatting (3DGS) also assumes a static illumination. Hence, we choose to not compare against 3DGS as these differences would overall limit the value of the comparison.
>
>
> ### Shadowing Already Handled by R3DGS
> > Weakness 1c
> As the reviewer correctly points out, the novelty here is not the handling of shadows but the way they are handled. Detailed shadowing effects can be predicted by our incident light prediction that is conditioned on the ray-traced visibility map on the 3D Gaussians.
> Compared to R3DGS we use a different light representation that models the changing illumination with explicit point light locations compared to the NeILF [37] based approach in R3DGS. Our approach together with the OLAT data enables higher frequency illumination and more direct control of the lighting compared to the static Spherical Harmonics based representation of R3DGS.
>
> ### Mask & Depth Input and Initialization Scheme
> > Weakness 3
> Our automatic mask generation only adds minimal overhead during preprocessing and generates masks based on either text prompts or control points (see section 4.1 real-world dataset for additional examples composited on black). Depth input is not needed for our pipeline. As we have a larger number of input images available the quality of the normals is sufficient without depth supervision. Normal estimation is additionally regularized through the deferred physically based renderer in agreement with [44]. Similarly, we don’t use the sparse point cloud from the SfM reconstruction as initialization (as is being done for R3DGS). Finally, we do not require a two staged training scheduling, we only use a single stage optimization initialized with random positions.
>
> ### Typos and Text Quality
> > Weakness 4
> Thanks for the pointers. We will fix the typos and rework the text for improved readability in section 1, 4.1 and 4.3.
>
>
> > 44: Ye et. al. 3D Gaussian Splatting with Deferred Reflection, arXiv 2024
> > 47: J. Schüle et al. Multi-Physical Tissue Modeling of a Human Urinary Bladder, EMBC 2021
> > 48: Yang et al. 3D reconstruction from endoscopy images: A survey, Computers in Biology and Medicine 2024

---

### Official Review · Reviewer_F9dx · 2024-07-10

**Soundness:** 3
**Presentation:** 2
**Contribution:** 2
**Rating:** 6
**Confidence:** 3

**Summary:**

This paper presents a method for recovering the shape and radiance transfer field (RTF) of an object from multi-view, OLAT data; placing an emphasis on translucent objects that exhibit subsurface scattering (SSS). In particular, the authors extend the framework of Relightable 3D Gaussian [Gao et al. 2024] in two ways: (1) instead of estimating per-Gaussian physically-based BRDF parameters, shading is performed in image space with a neural field that produces (potentially distinct) BRDF parameters for every pixel (backprojected into 3D); (2) in addition to predicting an incoming light field, an MLP predicts residual radiance due to SSS (i.e. a radiance transfer field that accounts for SSS alone), conditioned on 3D point, viewing direction, light source direction, and light source visibility.

In order to validate the efficacy of their framework, the authors collect a large multi-view OLAT dataset of translucent objects. With quantitative/qualitative comparisons on this dataset, the authors show favorable performance against existing state of the art works for shape and RTF estimation.

**Strengths:**

One strength of the work is the multi-view, OLAT dataset, which the authors say they plan to release. Containing 15 objects, with 100s of views and light positions (over 25000 images per object), this dataset seems like it took considerable effort to collect. I imagine it will prove quite useful in benchmarking approaches that perform shape/material estimation and relighting, esp. for translucent objects.

The quality of the results appears to be good (the qualitative results in the paper and supplement look reasonable), and quantitative results are strong compared to existing methods.

**Weaknesses:**

While the paper is, for the most part, well-structured and easy to follow, I found parts of the presentation to be slightly substandard. The pipeline figure is essentially just a collection of text boxes. A clearer delineation between network input, encoding, network architecture, network output (as well as an illustration of the purpose of each output), would help improve the clarity of this figure. I also don't think that the model design shown in this figure is fully accurate -- roughness, base color, and metalness should be position-dependent quantities only and should not depend on incoming/light direction.

I would also have liked it if the qualitative results were slightly better organized. I thank the authors for providing a number of renders of each object in the dataset under novel lighting/view. Collecting the results in a web page with labels for each video would have made the results easier to navigate.

I was surprised that so few visual comparisons to existing approaches are shown (Fig. 5 only).The KiloOSF results look far worse than I would expect -- the voxel artifacts seem to indicate a bug in evaluation/training.

Finally, while I don't doubt that the authors have built an effective system for shape and RTF recovery, I question the novelty of the contributions that enable this system. Performing deferred shading in pixel space for forward rendering/inverse rendering is hardly a new idea. Leveraging neural fields to predict RTFs (even residual RTFs) has also been done before in the literature (e.g. in Neural Fields for Structured Lighting [Shandilya et al. 2023]). Self-supervised visibility, the third listed contribution, is part of Relightable 3D Gaussian. Given, then, that the argument seems to be that a novel *combination* of these tools enables a better-performing system I would've hoped to see more comprehensive / higher quality evaluation.

**Questions:**

* Why do the results of KiloOSF look so bad compared to their results on their own dataset? What is the source of the voxel artifacts in the rendered images?
* Can you go into more detail on the conditioning of the outgoing SSS component and the incoming light? As written, it seems like the incoming light is conditioned on the outgoing view direction, which I don't think should be the case?

**Limitations:**

The authors adequately discuss the method's limitations and potential negative social impact.

---

> ### Author Rebuttal · Authors · 2024-08-06
>
> ### Improvements of Pipeline Overview (Fig. 2)
> > Weaknesses
> Thanks for the valuable feedback. We realize that the clarity of Fig. 2 in the paper can be improved. Please see Fig. 5 in the PDF for an updated version. We now tried to make clearer that base color, roughness and metalness are properties of each 3D Gaussian independent of incoming and light direction.
> See below as well as the paragraph on “Clearer Presentation of Key Idea and Contribution” in the global response for more information on the conditioning of the residual MLP.
>
> ### Conditioning of the SSS Residual and Incident Light Prediction
> > Question 2
> While the incident light is independent of the view direction, the residual SSS prediction might depend on it.
> In our proposed setup we jointly optimize the SSS Residual and incident light prediction.
> We find that a single lightweight MLP jointly conditioned on the 3D Gaussian parameters as well as the view direction and the ray traced visibility yields high quality results at interactive rates for the given task. See Tab. 2 “w/o Joint MLP” for comparison, where we split the MLP for residual and incident light inputting only relevant physical properties, leading to worse results. Our reasoning is that with this parameterization the main MLP is able to learn about the global light transport and therefore is capable of providing useful hints for the two output heads predicting the SSS Residual and the incident illumination, respectively. Both predictions are inherently constrained by the prediction of the incident light field which needs to drive the PBR rendering. However, the residual prediction can compensate for some limitations of the rendering models based on the provided input. As can be seen in the results in Fig. 4  the output is still a physically plausible material parameterization that enables multiple downstream applications like relighting or material editing.
>
> ### Combination of Tools
> > Weaknesses
> Our goal is to represent objects featuring translucent materials to be able to relight them or change individual material parameters. The focus is on enabling correct rendering of specular highlights and decomposition of the subsurface scattering component. This cannot be achieved with the existing methods and is a unique application at the time of writing.
> While we build on existing tools like 3DGS [15] and R3DGS [9] and ideas like deferred shading [11] we argue that our method is more than a combination of existing ideas.
> Our hybrid representation introduces a MLP for residual prediction of the subsurface scattering component which is constrained by the incident light prediction as architectural bias (as outlined above). While works like [46] show that MLPs can be used to represent the light transport also in complex scenes, our particular parameterization has not been proposed before as far as we know.
> Please also refer to “Clearer presentation of key idea and contribution” in the global section for further discussion of the residual prediction and MLP parameterization.
> Moreover, our work includes multiple modifications of the R3DGS framework like the support for OLAT illumination and the single stage optimization. The results in Tab. 2 also state that only the combination of all our proposed components achieve such high quality results.
> Finally, we also contribute a collection of OLAT scenes featuring translucent objects together with a preprocessing pipeline in the hope that this will enable follow up work in this field.
>
> ### More Comprehensive / Higher Quality Evaluation
> > Weaknesses
> Please find additional ablations in the newly added Fig. 3 and 4 as well as Tab. 2. We also demonstrate the editing applications our approach enables in more detail in Fig. 2. Please also refer to the global response on “Comprehensive evaluation” for more notes on the evaluation results. Given the novelty of the proposed editing applications there is no method to directly compare against. For a fair comparison we select KiloOSF [39] as the only real-time method that also enables relighting and novel view synthesis. We assume that the NeRF based methods can achieve similar quality, however at the cost of a much higher optimization and inference cost that makes comparisons on the full dataset unattractive.
>
> ### KiloOSF Results
> > Question 1
> While we did not directly analyze the voxel artifacts, we speculate that they are the results of the voxelization technique used in KiloNeRF and the complexity of the task gives the sparse views and light positions. The results shown in their paper [39] are produced by the default OSF NeRF approach training for multiple days and only achieving 0.27 FPS during inference, for fairness we choose to compare against their KiloOSF variant. There is no visual analysis shown for their real-time approach KiloOSF which achieves 14 FPS, that we could compare against. Therefore we don’t have a qualitative ground truth of the KiloOSF results. We utilized their official implementation and documentation at https://github.com/yuyuchang/KiloOSF and also saw similar artifacts using their own dataset that got worse when using non opaque objects. Our quantitative analysis shows similar results as provided within their work hinting these are the correct results.
>
> ### Webpage as Mode of Presentation
> > Weakness
> We thank the reviewer for the valuable feedback and taking the time to look into the supplementary material. We will launch a webpage including interactive viewing of the results together with the publication of the work and the dataset.
>
> > Again, we thank the reviewer for the valuable feedback. We hope that through our rebuttal it becomes clearer how our approach  uniquely enables accurate rendering and material parameterization for translucent objects as evidenced by new evaluations that will be added to the final paper.
>
>
> > 46: Shandilya et al. Neural Fields for Structured Lighting, ICCV 2023

---

> > ### Comment · Reviewer_F9dx · 2024-08-13
> >
> > I thank the authors for their rebuttal, and for answering many of my above questions. One confusion that I still have is the following: you state in your response here and to reviewer uYWZ that incoming light is conditioned on raytraced visibility. However, in the paper on line 196 you state that raytraced visibility is used to supervise a spherical harmonics-based visibility term v (similar to Relightable 3D Gaussian. Which of these is used to represent visibility / used to condition incoming lighting?

---

> > > ### Author Response · Authors · 2024-08-13
> > >
> > > We appreciate the reviewer’s thoughtful consideration of our rebuttal and apologize for any confusion. Both statements are indeed true. The incoming light is indirectly influenced by the visibility property of the Gaussians, as this property is provided as input to the MLP, which then outputs the incident light. Therefore, the incident light is conditioned on the ray-traced visibility. To supervise the visibility represented as spherical harmonics (SH), we ray-trace it similarly to R3DGS, by selecting random samples repeatedly during training for optimization. However, unlike R3DGS, our method does not require fine-tuning before training. Our neural representation is designed to be more flexible and compensating compared to the static representation of R3DGS, which assumes the sampled visibility as ground truth that directly influences the SH-represented incident light.

---

> > > > ### Comment · Reviewer_F9dx · 2024-08-13
> > > >
> > > > Thanks for your response. After reading through the rebuttal and author responses to reviewer questions, I am happy to upgrade my rating to a weak accept.

---

### Official Review · Reviewer_wfKX · 2024-07-11

**Soundness:** 4
**Presentation:** 2
**Contribution:** 3
**Rating:** 4
**Confidence:** 4

**Summary:**

This paper proposes the 3D Gaussian Splatting for subsurface scattering objects by decomposing the scene into subsurface scattering, diffuse and specular reflections, and object shape. By using multi-view OLAT (one light at a time) data of translucent objects, the proposed method optimizes BRDF parameters attributed to Gaussians and a small MLP that outputs subsurface scattering radiance and incident light. For accurate rendering, this paper proposes to incorporate deferred shading into 3DGS for specular highlights and explicitly handle shadowing by considering the visibility of Gaussians. The experimental results demonstrate the successful decomposition of the PBR parameters and the effectiveness of using 3DGS compared with the NeRF-based method.

**Strengths:**

+ The first method for handling subsurface scattering objects with 3DGS and enabling the decomposition of subsurface scattering, diffuse and specular reflections, and object shape of translucent objects.
+ Introduce deferred shading into 3DGS to enable accurate rendering of specular highlights.
+ Experimentally show that the proposed method successfully decomposes the PBR parameters and significantly outperforms the NeRF-based method for novel view synthesis, training time, and rendering speed.

**Weaknesses:**

- The representation of subsurface scattering is almost the same as the existing NeRF-based work [39], which takes points, viewing direction, and light direction as input of an MLP. The main difference is to simultaneously estimate incident light and use it for the physically-based rendering of specular and diffuse reflections like [9]. Although this difference improves the quality of rendering and enables the decomposition of PBR parameters, this method seems a combination of the existing works.
- The incident light $L_{in}$ can be view-dependent, which is physically inaccurate.

**Questions:**

- A clearer presentation of the key idea and its superiority beyond the just combination of the existing works (NeRF-based method and relightable 3DGS) is expected.
- Why does the MLP in Eq. 6 take the covariance and normal of a 3D Gaussian as input? The outputs may not depend on them.

**Limitations:**

Yes.

---

> ### Author Rebuttal · Authors · 2024-08-06
>
> ### Limited Novelty due to Recombination of Recent Techniques and Similarity to NeRF-based Approaches
> > Weakness 1 & Question 1
> To enable the first method for reconstruction of translucent objects that enables relighting and material editing we
> > - have performed multiple substantial modifications on the R3DGS baseline (Tab. 2),
> > - contribute a novel hybrid representation that has an architectural bias for decomposition and global light transport understanding,
> > - created and will release an open dataset of OLAT scenes featuring diverse translucent objects.
> >
> >Compared to the NeRF based approaches [42, 39] our hybrid representation features explicit surfaces modeled by the 3D Gaussians and a novel parameterization of a global MLP for the volumetric part.
> Our reasoning for the residual setup is presented in “Clearer Presentation of Key Idea and Contribution” of the global response.
> The key insight here is that a single MLP backbone together with two heads for volumetric residual and incident illumination prediction, respectively, performs best in our data setting (Tab. 2). Please also refer to the following paragraphs on the parameterization of the MLP and the incident light prediction. We will rework the relevant sections in the paper to make this reasoning clearer.
> In addition to the speed up that 3DGS [15] brings through point based rasterization, we identify the advantage of the strong surface prior from 3DGS for our decomposition task as a key difference compared to existing NeRF based approaches. Our formulation of a surface based shading with a volumetric residual to represent the subsurface scattering component is well suited for many editing applications and leads to a better geometry reconstruction compared to NeRF based approaches as well as a subsurface scattering representation much closer to the physical model than the plain R3DGS [9] (see the examples in section 4.2 comparison of the paper and the new ablation results as part of this rebuttal, specifically Fig. 4 and Tab. 2 and 3).
> We find that R3DGS’s capability to render specular reflections is limited by the size of the 3D Gaussians and find a straightforward solution by using a deferred shading pipeline.
> Furthermore, we adapt the framework for the OLAT setting (also see “Choice of OLAT Data and Relighting” in the global response), replacing the global environment map and the local illumination representation with a point light based representation. This enables higher frequency illumination and more direct control of the lighting compared to the Spherical Harmonics based representation of R3DGS.
> Finally we abandon the two stage optimization approach from R3DGS and propose a single optimization stage without the need of initialization with a sparse point cloud that might be hard to obtain for translucent objects.
> Since we could not find a suitable open dataset we also contribute a collection of OLAT scenes with translucent objects as part of this work in the hope that this will enable follow up work in this field.
>
> ### Parameterization of MLP (Eq. 6)
> > Question 2
> Eq. 6 accurately reflects the implementation, meaning that the MLP does indeed take covariance (rotation and scale parameter) and normal of a 3D Gaussian as input. The original intent was to condition the MLP on all surface information we have from the 3D Gaussian and to possibly receive an additional gradient update on the normal parameter.
> Inspired by the reviewer’s comment we ran additional experiments and it turns out that conditioning without normals and covariance  parameters is sufficient to achieve the reported quality.
>
> ### Incident Light Prediction is Physically Inaccurate
> > Weakness  2
> Thanks to the reviewer’s feedback we realize that the notation $L_{in}$ in Equation 6 in the paper might be a bit misleading in the context of physically-based rendering. We will update the section in the paper to make it clearer that $L_{in}$ is a prediction of our model that is close to the true physical quantity but might be slightly offset to compensate for limitations in the model.
> While the incident light is independent of the view direction, the residual prediction is not. Using a single MLP backbone together with two heads for volumetric residual and incident illumination prediction, respectively, performs best in our experiments (see Tab. 2 row “w/o Joint MLP”). The hypothesis here is that the processing inside the MLP includes reasoning about the global light transport inside the object that is inherently constrained by the prediction of the incident light field (which needs to work for the PBR renderer). By relaxing the strictly physical model a little we gain better quality output, potentially compensating limitations of our rendering model.
> Please also see the global response on “Clearer Presentation of Key Idea and Contribution” for further notes.
>
> > We want to thank the reviewer again for the thorough analysis and hope that the key ideas of our method became clearer through this rebuttal. As mentioned above we will rework the introduction and method section of the paper to reflect the reviewer’s feedback. We will also add the additional analysis provided in the accompanying PDF to the supplementary section. We invite the reviewer to also take a look into it as it illustrates the effects of the explanations above.

---

> > ### Comment · Reviewer_wfKX · 2024-08-12
> >
> > I appreciate your response. Although I acknowledge some improvements from 3DGS and NeRF-based methods and they realize the first 3DGS-based method for reconstruction and decomposition of translucent objects, these improvements seem engineering works rather than novel ideas. Thus, I will keep my rating.

---

> > > ### Author Response · Authors · 2024-08-13
> > >
> > > We appreciate the reviewer’s feedback. While it is true that our approach builds upon existing 3DGS and NeRF-based methods, we would like to emphasize once more that the novelty lies in our analytical approach combining these methods to achieve such high quality results, as demonstrated in the Ablation (Rebuttal Tbl. 2) when compared to the baseline and further in comparison to previous methods (Paper Tbl. 1). While some of these findings certainly involve engineering effort, we want to highlight that the joint prediction of residual and incident light, which brings about a drastic performance increase through this specific parameterization, has not been proposed before. Furthermore, regarding the deferred shading, we provide an analytical explanation for why specular details in previous works, including NeRF-based methods for translucent objects, are underrepresented and offer a solution to this issue. Also the constructed dataset is a novel contribution to the field of translucent object reconstruction.
> > >
> > > We want to stress that all these contributions are not only technically challenging but also offer new insights into this underrepresented field of research, enabling new capabilities that were previously unattainable

---

### Official Review · Reviewer_pNBB · 2024-07-12

**Soundness:** 3
**Presentation:** 3
**Contribution:** 2
**Rating:** 6
**Confidence:** 4

**Summary:**

This paper proposes an algorithm to reconstruct relightable objects with subsurface scattering (SSS) effects. It proposes to model SSS as a residual to surface PBR using a neural network. It also proposes to perform shading in image space (i.e. deferred shading) to improve specularity. Lastly, the SSS network takes ray-trace light visibility as input, accounting for shadowing effects for the SSS component.

**Strengths:**

1. Modeling SSS is an under-explored research area in inverse rendering, and the demonstrated results in this paper look promising.

2. Using a neural network to predict residuals to surface-based PBR is straightforward yet effective, as demonstrated in the paper.

**Weaknesses:**

1. The approach assumes known point-light, which limits its applicability in real-world scenarios

2. The novelty is overclaimed: GS-IR ([19] in the paper) has already introduced deferred/pixel-space shading to Gaussian splatting. There are also several concurrent works reaching the same conclusion as this paper (e.g. deferred shading improves specularity) [1, 2], please cite them and discuss the relations between them and this paper in the final version of this paper. However, I do acknowledge that this paper is one of the first papers that argues deferred shading improves specularity.

3. Insufficient ablation: only qualitative results on ablation are presented. It would be preferable to have quantitative results on the ablation baselines. Also, the comparison to relightable 3D Gaussians is not fair. A more fair comparison is to remove the SSS part of the proposed method, which becomes a variant of relightable 3D Gaussians that can handle known illumination.


[1] Ye et. al. 3D Gaussian Splatting with Deferred Reflection, arXiv 2024

[2] Wu et. al. DeferredGS: Decoupled and Editable Gaussian Splatting with Deferred Shading, arXiv 2024

**Questions:**

1. What prevents the method from working without knowing point-light?

2. Do the results in Tab. 1 include novel lighting conditions? How are light intensity set in both synthetic and real dataset? How would the method work if the light intensity changes drastically, i.e. can the model handle light intensity changes from e.g. 5 (training) to 20 (testing)?

3. Only point light is demonstrated in the results. Would the model still work under natural illumination (e.g. image-based lighting)? According to Eq. 6 it seems to be designed specifically for a single point light, which is quite limiting. Would be good to see both qualitative and quantitative results on synthetic datasets with image-based lighting

**Limitations:**

Limitation is discussed, though I think a crucial part (not supporting image-based lighting) is missing. Please see questions.

---

> ### Author Rebuttal · Authors · 2024-08-06
>
> ### Optimization under Unknown Illumination
> > Question 1
> Optimization with unknown point light locations adds additional dimensions to an already under-constrained optimization problem which we consider out of scope of this work as we focus on the geometry representation and rendering part.
> As of interest for multiple reviewers we discussed the advantages of OLAT data for the reconstruction of subsurface scattering (SSS) in the global response. Opening up the illumination to more complex patterns and natural illumination eventually is an interesting research trajectory that we plan to pursue in future work. To successfully reconstruct the subsurface scattering component of the light transport it is necessary to understand the global light transport in the scene including the heterogeneous volume structure of an object. In theory our method supports any representation of the incident light field. Assuming an unknown illumination this light field would need to be estimated in addition to the light transport, though. This adds additional complexity to the optimization problem. New priors on either illumination or geometry would be needed here which could be potentially obtained with the help of our acquired dataset. However, given the overall complexity of the problem we decided to focus on the geometry and material representation aspect first and consider improvements of the illumination representation beyond the incident light prediction as out of the scope of this work.
> Still, our work is robust to deviations from the actual light position and potentially could also optimize an offset on the point light positions without much additional effort.
>
> ### Concurrent Works on Deferred Shading for 3DGS
> > Weakness 2
> We thank the reviewer for hinting at existing deferred shading techniques in the realm of Gaussian Splatting and we will gladly incorporate these in the related work section. The motivation to use a deferred shading pipeline is the limited capability of the shading approach of R3DGS [9] to reproduce specular highlights. See Fig. [3] for a visual comparison.
> While Ye et al. [44] and Wu et al. [45] focus on the blending and propagation of normal directions between overlapping 3D Gaussians, our analysis based on the 3D Gaussian areas adds to the understanding of these effects. [45] train a SDF in parallel to improve the surface geometry which we do not need to. Also our shading model differs from the existing ones as we add the explicit SSS Residual term to the PBR material parameters and predict detailed incident light.
>
>
> ### Novel Lighting in Tab. 1
> > Question 2
> The values shown in Tab. 1 (Paper) include novel lighting positions, we build a train and test set with a 50/50 split for the real world and synthetic dataset (see Sec. 4.1 Paper). Still, these light positions are part of the lightstage, therefore, have the same distance to the object. For truly new light positions we would like to refer to the videos in the supplementary material and Fig. 2 in the rebuttal. Varying intensity or colored light was not available within the test set but we now added a comparison for the synthetic scenes in Tab. 1 and Fig. 1. Illumination editing is still possible as described next.
>
> ### Illumination Editing including Light Intensity
> > Question 3
> In Fig. 2 we present various editing modes our method enables. Light intensity can be changed by scaling the incident light field and residual prediction or the linear output of the deferred shading stage. As we only train with a fixed light spectrum and intensity extreme edits will result in inaccurate results for the residual component, though. Extending the setting to a spectral subsurface scattering model e.g. by training on different spectral responses in addition to the white lights is an interesting idea for future work. We thank the reviewer for the inspiring comment.
>
> ### Image Based Lighting
> > Question 3
> Indeed our method is capable of achieving image based lighting. As this was of interest for multiple reviewers we would like to refer to the global comment as well as Fig. 1 for qualitative and Tab. 1 for quantitative results. As can be seen the relighting results closely resemble the ground truth renders regarding color, light direction and subsurface scattering. Note, that the PSNR metric is affected by denoising artifacts and the noise residual from the Monte-Carlo path tracing in the ground truth data.
>
> ### Insufficient Ablation
> > Weakness 3
> As introduced in the global response on “Comprehensive Evaluation and Ablation” we now provide qualitative and quantitative results of multiple ablations. In Tab. 2  and Fig. 3 can be seen that the residual prediction, the PBR rendering and the deferred shading component all contribute to the quality of the final results. Our model is capable of representing the SSS component and specular materials and also enables intuitive editing of the material parameters as shown in Fig. 2.
>
> ### Comparison against Relightable 3D Gaussian Splatting (R3DGS)
> > Weakness 3
> In Tab. 2 we also compare against a variant of R3DGS [9] with known illumination. Please note that the original R3DGS approach only supports a single illumination setting, therefore differing from our OLAT setting and not capable of learning to disentangle SSS and surface reflection.
> Compared to R3DGS we use a different light representation that models the changing illumination with explicit point light locations compared to the NeILF [37] based approach in R3DGS. The results in Tab. 2 clearly indicate the limitations of the base model in our experiment setting.  Our approach together with the OLAT data enables higher frequency illumination and more direct control of the lighting compared to the Spherical Harmonics based representation of R3DGS.
>
> > 44: Ye et. al. 3D Gaussian Splatting with Deferred Reflection, arXiv 2024
> > 45: Wu et. al. DeferredGS: Decoupled and Editable Gaussian Splatting with Deferred Shading, arXiv 2024

---

> ### Comment · Reviewer_pNBB · 2024-08-12
>
> I think the authors addressed one of my major concerns which is image-based lighting. Though it nullifies the initial argument that the model can achieve real-time rendering. It would make the paper much stronger if the method could achieve real-time rendering for image-based lighting
>
> The added ablations are also sufficient.
>
> I agree that unknown natural illumination + SSS is a challenging setup, as the community hasn't even solved opaque objects reconstruction problem yet. The newly constructed dataset does seem to be a good contribution to the community.
>
> On the other hand, I concur with other reviewers that the technical novelty of the work is quite limited. Judging by all these factors I would slightly increase my score but wouldn't argue strongly for acceptance if other reviewers don't agree.

---

> > ### Author Response · Authors · 2024-08-13
> >
> > We are grateful that we could address the reviewer’s major concern and clarify open questions. The reviewer is correct that a real-time evaluation of image-based lighting (IBL) was not shown. However, we want to emphasize that our method still significantly outperforms previous NeRF-based methods in terms of speed for IBL evaluation. Additionally, as we highlighted, with pre-computation, the IBL can be independently calculated very quickly. The numbers we provided represent naive sequential computation, which could be further optimized through parallelization to definitely achieve real-time evaluation. We thank the reviewer for his positiv feedback on our added ablations and the contribution of our newly constructed dataset.

---

### Author Rebuttal · Authors · 2024-08-06

R1 pNBB - R2 wfKX - R3 F9dx - R4 uYWZ - R5 uk4p
Find cited works in the original paper, Fig. numbers refer to rebuttal PDF

We thank the reviewers for their constructive feedback and for recognizing our effort to advance an “under-explored research area in inverse rendering” [R1].

We propose a method to “reconstruct relightable objects with subsurface scattering (SSS) effects” [R1] by “decomposing the scene into subsurface scattering, diffuse and specular reflections, and object shape” [R2]. This decomposition in combination with the Gaussian Splatting (GS) “framework enables material editing, relighting, and novel view synthesis in real-time” [R4].
We appreciate the acknowledgement of the reviewers, that the “proposed method successfully decomposes the PBR parameters and significantly outperforms the NeRF-based method for novel view synthesis, training time, and rendering speed” [R2]. Further, that the “quantitative results are strong compared to existing methods” [R3], overall achieving “comparable results and fast rendering speed” [R4]. Additionally, the created dataset can be a “great contribution to the vision and graphics community” [R5], enabling e.g. benchmarking of “approaches that perform shape/material estimation and relighting, esp. for translucent objects” [R3].

We hope to clarify open questions and show new insights to our method (please also see attached PDF) that underline its contribution to the research field. In the following sections we address questions and comments raised by multiple reviewers.

## Clearer Presentation of Key Idea and Contribution [R2, R3]
To the best of our knowledge the proposed method is the first to show joint reconstruction and decomposition of translucent objects for high quality relighting and material editing in real-time.
We will rework the introduction section of the paper to make the key components of our contribution clearer as outlined below.
At the core we propose a hybrid representation that extends 3D Gaussian Splatting with PBR material parameters [9] and deferred shading [11] with a light-weight residual prediction network to learn shading components not modeled by the surface shader. We focus on the subsurface scattering component for translucent objects which is underrepresented in recent research on neural inverse rendering.
We constrain the network predicting the outgoing subsurface scattering (SSS) radiance by jointly predicting the incident radiance used for the PBR rendering step. The incident light prediction is effectively regularized as it is used in a physically based renderer together with the independently optimized material parameters. We highlight this in Tab. 2 of our newly added ablations showing the difference in performance between a joint MLP and two independent ones. By relaxing the physical definition of the residual prediction a little we gain better quality output, potentially compensating limitations of our rendering model.

## Choice of OLAT Data and Relighting [R1, R4]
One Light at A Time (OLAT) data makes it possible to disentangle the subsurface scattering effects and reflectance at the surface that would need additional priors otherwise.
Inverse rendering of translucent objects is a severely ill-posed problem. The OLAT setting is common here [42, 39] as it helps to recover the complex global lighting in the scene by providing an impulse response of the system; every image in the dataset is only illuminated by a single light of known position. This leads to a more accurate reproduction of highlights and SSS than if illuminated by an environment light. We acknowledge the limitations such an acquisition setup imposes and, therefore, want to make our acquired data available. Once reconstructed, our model can work in arbitrary lighting settings.

## Image Based Relighting [R1, R3, R4, R5]
In Fig. 1 we show that our method achieves image based lighting with high visual quality. First we sample the HDR maps representing distant environment illumination. The samples don’t need to correspond to OLAT samples used in training and could be placed much denser. Using this approach we can generate a relit frame in about 20 seconds for a medium resolution. To speed up generation we can precompute a reflectance field assuming white light. We compute the relit view as the sum over the reflectance field scaled by the environment illumination before applying tone mapping for display. This runs in a fraction of a second. As can be seen in Fig. 1 an object can be rendered in different illumination settings yielding consistent photorealistic results.

## Editing Applications [R4, R5]
We demonstrate the editing applications our approach enables in more detail in Fig. 2. We show:
- Relighting with environment map based lighting
- Single light illumination outside the training domain
- Color changes of base color or SSS Residual
- Editing of material parameters to make the appearance more metallic, shinier or rougher
- Changing opacity and intensity of the SSS

Further, our SSS Residual adds editing capabilities that are not available in previous methods [9].

## Comprehensive Evaluation and Ablation [R1, R3, R4]
Please find the additional ablations in Tab. 2 and visual examples in Fig. 3. The provided evidence underlines the pipeline choices presented in our method. Please note that the metrics used only have limited capability to capture the fine details in visual quality that the improved representation of specular highlights adds (see Fig. 3). Compared to previous methods we allow for detailed editing of the reflectance parameters at the surface independently of the volumetric residual while keeping and extending the relighting functionality. The analysis of the material prediction and the intermediate shading buffers highlights the physical plausibility of our results (see Fig. 4).

We would like to thank all reviewers again for their valuable feedback that already led to an improved submission.

---

### Decision · Program_Chairs · 2024-09-25

**Decision:**

Accept (poster)

**Comment:**

This paper received 5 mixed, but positive leaning reviews ---- three 6s (weak accept), one 5 (borderline accept), one 4 (borderline reject).

There was general consensus around the novelty of problem being addressed (incorporating subsurface scattering in 3DGS for volumetric reconstruction of translucent objects), the quality of results and the usefulness of the proposed dataset.

There were several concerns raised - from evaluation and experimental standpoints, and also regarding technical novelty. The authors provided thorough responses which assuaged the evaluation concerns to a large extent. One of the reviewers had lingering doubts about the technical novelty arguing that the paper mostly builds upon existing approaches. On the other hand, most other reviewers appreciated the principled approach the paper takes for handling subsurface scattering.

Given these, on balance, an accept decision was reached.

The authors are strongly encouraged to incorporate the reviewer comments -- especially around new validation content that was promised in the rebuttal -- in the camera-ready version.